# Is Value Learning Really
# the Main Bottleneck in Offline RL?

**Seohong Park**[1]    **Kevin Frans**[1]    **Sergey Levine**[1]    **Aviral Kumar**[2]
[1]University of California, Berkeley    [2]Carnegie Mellon University
seohong@berkeley.edu

## Abstract

While imitation learning requires access to high-quality data, offline reinforcement learning (RL) should, in principle, perform similarly or better with substantially lower data quality by using a value function. However, current results indicate that offline RL often performs worse than imitation learning, and it is often unclear what holds back the performance of offline RL. Motivated by this observation, we aim to understand the bottlenecks in current offline RL algorithms. While poor performance of offline RL is typically attributed to an imperfect value function, we ask: *is the main bottleneck of offline RL indeed in learning the value function, or something else?* To answer this question, we perform a systematic empirical study of (1) value learning, (2) policy extraction, and (3) policy generalization in offline RL problems, analyzing how these components affect performance. We make two surprising observations. **First**, we find that the choice of a policy extraction algorithm significantly affects the performance and scalability of offline RL, often more so than the value learning objective. For instance, we show that common value-weighted behavioral cloning objectives (*e.g.*, AWR) do not fully leverage the learned value function, and switching to behavior-constrained policy gradient objectives (*e.g.*, DDPG+BC) often leads to substantial improvements in performance and scalability. **Second**, we find that a big barrier to improving offline RL performance is often imperfect policy generalization on test-time states out of the support of the training data, rather than policy learning on in-distribution states. We then show that the use of suboptimal but high-coverage data or test-time policy training techniques can address this generalization issue in practice. Specifically, we propose two simple test-time policy improvement methods and show that these methods lead to better performance.

## 1 Introduction

Data-driven approaches that convert offline datasets of past experience into policies are a predominant approach for solving control problems in several domains [9, 49, 51]. Primarily, there are two paradigms for learning policies from offline data: imitation learning and offline reinforcement learning (RL). While imitation requires access to high-quality demonstration data, offline RL loosens this requirement and can learn effective policies even from suboptimal data, which makes offline RL preferable to imitation learning in theory. However, recent results show that tuning imitation learning by collecting more expert data often outperforms offline RL even when provided with sufficient data in practice [36, 48], and it is often unclear what holds back the performance of offline RL.

The primary difference between offline RL and imitation learning is the use of a *value function*, which is absent in imitation learning. The value function drives the learning progress of offline RL methods, enabling them to learn from suboptimal data. Value functions are typically trained via temporal-difference (TD) learning, which presents convergence [40, 55] and representational [27, 29, 56] pathologies. This has led to the conventional wisdom that the gap between offline RL and imitation is a direct consequence of poor value learning [26, 33, 36]. Following up on this conventional wisdom, recent research in the community has been devoted towards improving the value function quality of offline RL algorithms [1, 11, 14, 19, 25, 26]. While improving value functions will definitely help improve performance, we question whether this is indeed the best way to maximally improve

the performance of offline RL, or if there is still headroom to get offline RL to perform better even with current value learning techniques. More concretely, given an offline RL problem, we ask: *is the bottleneck in learning the value function, the policy, or something else? What is the best way to improve performance given the bottleneck?*

We answer these questions via an extensive empirical study. There are three potential factors that could bottleneck an offline RL algorithm: (**B1**) imperfect **value** function estimation, (**B2**) imperfect **policy** extraction guided by the learned value function, and (**B3**) imperfect policy **generalization** to states that it will visit during evaluation. While all of these contribute in some way to the performance of offline RL, we wish to identify how each of these factors interact in a given scenario and develop ways to improve them. To understand the effect of these factors, we use data size, quality, and coverage as levers for systematically controlling their impacts, and study the "data-scaling" properties, *i.e.*, how data quality, coverage, and quantity affect these three aspects of the offline RL algorithm, for three value learning methods and three policy extraction methods on diverse types of environments. These data-scaling properties reveal how the performance of offline RL is bottlenecked in each scenario, hinting at the most effective way to improve the performance.

Through our analysis, we make two surprising observations, which naturally provide actionable advice for both domain-specific practitioners and future algorithm development in offline RL. **First, we find that the choice of a *policy extraction* algorithm often has a larger impact on performance than value learning algorithms**, despite the policy being subordinate to the value function in theory. This contrasts with the common practice where policy extraction often tends to be an afterthought in the design of value-based offline RL algorithms. Among policy extraction algorithms, we find that behavior-regularized policy gradient (*e.g.*, DDPG+BC [14]) almost always leads to much better performance and favorable data scaling than other widely used methods like value-weighted regression (*e.g.*, AWR [46, 47, 58]). We then analyze why constrained policy gradient leads to better performance than weighted behavioral cloning via extensive qualitative and quantitative analyses.

**Second, we find that the performance of offline RL is often heavily bottlenecked by how well the policy *generalizes* to *test-time* states, rather than its performance on training states.** Namely, our analysis suggests that existing offline algorithms are often already great at learning an optimal policy from suboptimal data on *in-distribution* states, to the degree that it is saturated, and the performance is often simply bottlenecked by the policy accuracy on novel states that the agent encounters at test time. This provides a new perspective on *generalization* in offline RL, which differs from the previous focus on pessimism and behavioral regularization. Based on this observation, we provide two practical solutions to improve the generalization bottleneck: the use of high-coverage datasets and test-time policy extraction techniques. In particular, we propose new on-the-fly policy improvement techniques that further distill the information in the value function into the policy on test-time states *during evaluation rollouts*, and show that these methods lead to better performance.

Our main contribution is an analysis of the bottlenecks in offline RL as evaluated via data-scaling properties of various algorithmic choices. Contrary to the conventional belief that value learning is the bottleneck of offline RL algorithms, we find that the performance is often limited by the choice of a policy extraction objective and the degree to which the policy generalizes at test time. This suggests that, with an appropriate policy extraction procedure (*e.g.*, gradient-based policy extraction) and an appropriate recipe for handling generalization (*e.g.*, test-time training with the value function), collecting more high-coverage data to train a value function is a universally better recipe for improving offline RL performance, whenever the practitioner has access to collecting some new data for learning. These results also imply that more research should be pursued in developing policy learning and generalization recipes to translate value learning advances into performant policies.

## 2   Related work

Offline reinforcement learning [31, 33] aims to learn a policy solely from previously collected data. The central challenge in offline RL is to deal with the distributional shift in the state-action distributions of the dataset and the learned policy. This shift could lead to catastrophic value overestimation if not adequately handled [33]. To prevent such a failure mode, prior works in offline RL have proposed diverse techniques to estimate more suitable value functions solely from offline data via conservatism [8, 26], out-of-distribution penalization [14, 53, 59], in-sample maximization [17, 25, 61], uncertainty minimization [1, 19, 60], convex duality [32, 41, 50], or contrastive learning [11]. Then, these methods train policies to maximize the learned value function with behavior-regularized policy gradient (*e.g.*, DDPG+BC) [14, 34], weighted behavioral cloning (*e.g.*, AWR) [46, 47], or sampling-based action selection (*e.g.*, SfBC) [7, 15, 21]. Depending on the algorithm, these value

learning and policy extraction stages can either be interleaved [14, 26, 42] or decoupled [5, 11, 17, 25]. Despite the presence of a substantial number of offline RL algorithms, relatively few works have aimed to analyze and understand the practical challenges in offline RL. Instead of proposing a new algorithm, we mainly aim to understand the current bottlenecks in offline RL via a comprehensive analysis of existing techniques so that we can inform future methodological development.

Several prior works have analyzed individual components of offline RL or imitation learning algorithms: value bootstrapping [14, 15], representation learning [27, 29, 62], data quality [4], differences between RL and behavioral cloning (BC) [28], and empirical performance [10, 23, 35, 36, 54]. Our analysis is distinct from these lines of work: we analyze challenges appearing due to the interaction between these individual components of value function learning, policy extraction, and generalization, which allows us to understand the bottlenecks in offline RL from a *holistic* perspective. This can inform how a practitioner could extract the most by improving one or more of these components, depending upon their problem. Perhaps the closest study to ours is Fu et al. [13], which study whether representations, value accuracy, or policy accuracy can explain the performance of offline RL. While this study makes insightful recommendations about which algorithms to use and reveals the potential relationships between some metrics and performance, the conclusions are only drawn from D4RL locomotion tasks [12], which are known to be relatively simple and saturated [48, 53], and the data-scaling properties of algorithms are not considered. In addition, this prior study does not identify policy *generalization*, which we find to be one of the most substantial yet overlooked bottlenecks in offline RL. In contrast, we conduct a large-scale analysis on diverse environments (*e.g.*, pixel-based, goal-conditioned, and manipulation tasks) and analyze the bottlenecks in offline RL with the aim of providing actionable takeaways that can enhance the performance and scalability of offline RL.

## 3 Main hypothesis

Our primary goal is to understand when and how the performance of offline RL can be bottlenecked in practice. As discussed earlier, there exist three potential factors that could bottleneck an offline RL algorithm: (**B1**) imperfect **value** function estimation from data, (**B2**) imperfect **policy** extraction from the learned value function, and (**B3**) imperfect **generalization** on the test-time states that the policy visits in evaluation rollouts. We note that the bottleneck of an offline RL algorithm under a certain dataset can always be attributed to one or some of these factors, since the policy will attain optimal performance if both value learning and policy extraction are perfect, and perfect generalization to test-time states is possible.

**Our main hypothesis** in this work is that, somewhat contrary to the prior belief that the accuracy of the value function is the primary factor limiting performance of offline RL methods, **policy learning is often the main bottleneck of offline RL**. In other words, while value function accuracy is certainly important, how the policy is extracted from the value function (**B2**) and how well the agent generalizes on states that it visits at the deployment time (**B3**) are often the main factors that significantly affect both the performance and scalability of offline RL. To verify this hypothesis, we conduct two main analyses in this paper. In Section 4, we compare the effects of value learning and policy extraction on performance under various types of environments, datasets, and algorithms (**B1** and **B2**). In Section 5, we analyze the degree to which the policy generalizes on test-time states affects performance (**B3**).

## 4 Empirical analysis 1: Is it the value or the policy? (**B1** and **B2**)

We first perform controlled experiments to identify whether imperfect value functions (**B1**) or imperfect policy extraction (**B2**) contribute more to holding back the performance of offline RL in practice. To systematically compare value learning and policy extraction, we run different algorithms while varying the *the amounts of data* for value function training and policy extraction, and draw **data-scaling matrices** to visualize the aggregated results. Increasing the amount of data provides a convenient lever to control the effect of each component, enabling us to draw conclusions about whether the value or the policy serves as a bigger bottleneck in different regimes when different amounts of training data are available (or can be collected by a practitioner for a given problem), and to understand the differences between various algorithms.

To clearly dissect value learning from policy learning, we focus on offline RL methods with *decoupled* value and policy training phases (*e.g.*, One-step RL [5], IQL [25], CRL [11], etc.), where policy learning does not affect value learning. In other words, we focus on methods that first train a value function without involving policies, and then extract a policy from the learned value function with a separate objective. While this might sound a bit restrictive, we surprisingly find that policy learning

is often the main bottleneck *even in these decoupled methods*, which attempt to solve a simple, single-step optimization problem for extracting a policy given a static and stationary value function.

## 4.1 Analysis setup

We now introduce the value learning objectives, policy extraction objectives, and environments that we study in our analysis (see Appendix B for preliminaries).

**Value learning objectives.** We consider three decoupled value learning objectives that fit value functions without involving policy learning: **(1) implicit Q-learning (IQL)** [25], **(2) SARSA** [5], and **(3) contrastive RL (CRL)** [11]. IQL fits an optimal Q function ($Q^*$) by approximating the Bellman optimality operator with an expectile loss. SARSA fits a behavioral Q function ($Q^\beta$) using the Bellman evaluation operator. In goal-conditioned tasks, we employ CRL instead of SARSA, which similarly fits a behavioral Q function, but with a different contrastive learning-based objective that leads to better performance. We refer to Appendix D.1 for detailed descriptions of these value learning methods.

**Policy extraction objectives.** Prior works in offline RL typically use one of the following objectives to extract a policy from the value function. All of them are built upon the same principle: maximizing values while being close to the behavioral policy, to avoid the exploitation of erroneous critic values.

- **(1) Weighted behavioral cloning (*e.g.*, AWR).** Weighted behavioral cloning is one of the most widely used offline policy extraction objectives for its simplicity [25, 42, 44, 46, 47, 58]. Among weighted behavioral cloning methods, we consider advantage-weighted regression (AWR [46, 47]) in this work, which maximizes the following objective:

$$\max_\pi \ \mathcal{J}_{\mathrm{AWR}}(\pi) = \mathbb{E}_{s,a\sim\mathcal{D}}[e^{\alpha(Q(s,a)-V(s))} \log \pi(a \mid s)], \tag{1}$$

  where $\alpha$ is an (inverse) temperature hyperparameter. Intuitively, AWR assigns larger weights to higher-advantage transitions when cloning behaviors, which makes the policy selectively copy only good actions from the dataset.

- **(2) Behavior-constrained policy gradient (*e.g.*, DDPG+BC).** Another popular policy extraction objective is behavior-constrained policy gradient, which directly maximizes Q values while not deviating far away from the behavioral policy [1, 14, 19, 26, 59]. In this work, we consider the objective that combines deep deterministic policy gradient and behavioral cloning (DDPG+BC [14]):

$$\max_\pi \ \mathcal{J}_{\mathrm{DDPG+BC}}(\pi) = \mathbb{E}_{s,a\sim\mathcal{D}}[Q(s,\mu^\pi(s)) + \alpha \log \pi(a \mid s)], \tag{2}$$

  where $\mu^\pi(s) = \mathbb{E}_{a\sim\pi(\cdot|s)}[a]$ and $\alpha$ is a hyperparameter that controls the strength of the BC regularizer.

- **(3) Sampling-based action selection (*e.g.*, SfBC).** Instead of learning an explicit policy, some previous methods implicitly define a policy as the action with the highest value among action samples from the behavioral policy [7, 15, 18, 21]. In this work, we consider the following objective that selects the $\arg\max$ action from behavioral candidates (SfBC [7]):

$$\pi(s) = \underset{a\in\{a_1,\ldots,a_N\}}{\arg\max} \ [Q(s,a)], \tag{3}$$

  where $a_1,\ldots,a_N$ are sampled from the learned BC policy $\pi^\beta(\cdot \mid s)$ [7, 21].

**Environments and datasets.** To understand how different value learning and policy extraction objectives affect performance and data scalability, we consider eight environments (Figure 10) across state- and pixel-based, robotic locomotion and manipulation, and goal-conditioned and single-task settings with varying levels of data suboptimality: **(1)** `gc-antmaze-large`, **(2)** `antmaze-large`, **(3)** `d4rl-hopper`, **(4)** `d4rl-walker2d`, **(5)** `exorl-walker`, **(6)** `exorl-cheetah`, **(7)** `kitchen`, and **(8)** `gc-roboverse`. We highlight some features of these tasks: `exorl-{walker, cheetah}` are tasks with highly suboptimal, diverse datasets collected by exploratory policies, `gc-antmaze-large` and `gc-roboverse` are goal-conditioned ('gc-') tasks, and `gc-roboverse` is a *pixel-based* robotic manipulation task with a $48 \times 48 \times 3$-dimensional observation space. For some tasks (*e.g.*, `gc-antmaze-large` and `kitchen`), we additionally collect data to enhance dataset sizes to depict scaling properties clearly. We refer to Appendix D.2 for the complete task descriptions.

## 4.2 Results: Policy extraction mechanisms substantially affect data-scaling trends

Figure 1 shows the data-scaling matrices of three policy extraction algorithms (AWR, DDPG+BC, and SfBC) and three value learning algorithms (IQL and {SARSA or CRL}) on eight environments, aggregated from a total of 15,488 runs (8 seeds for each cell, numbers after "±" denote standard

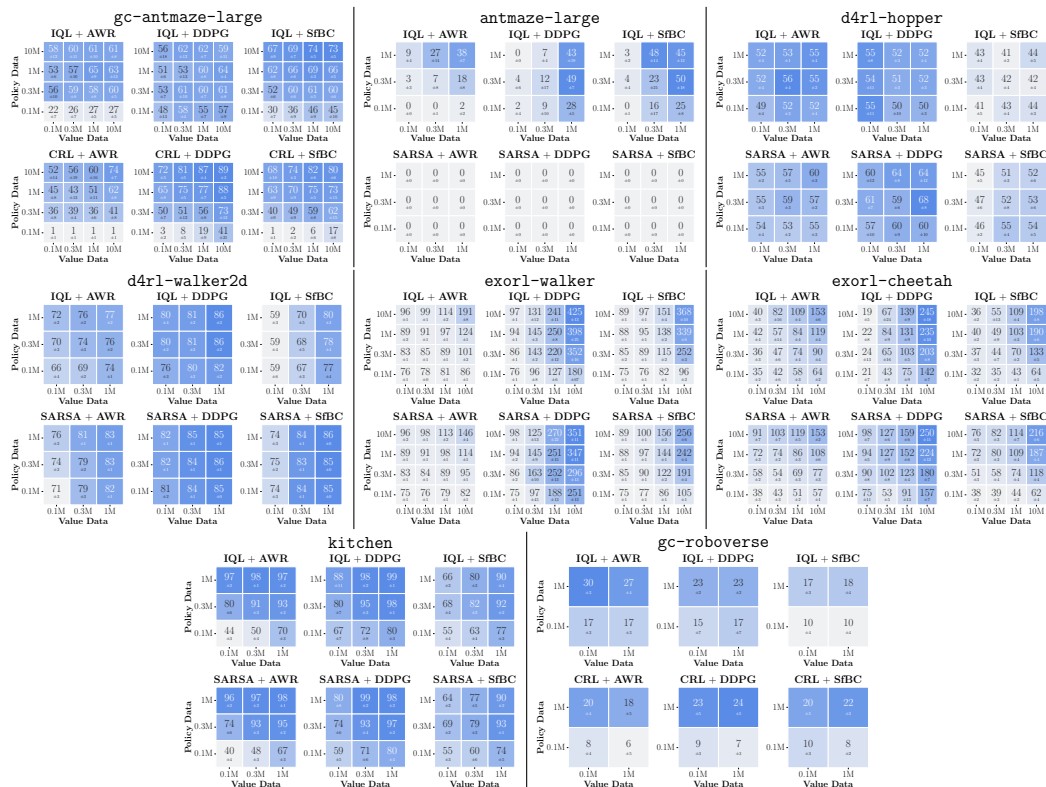

Figure 1: **Data-scaling matrices of three policy extraction methods (AWR, DDPG+BC, and SfBC) and three value learning methods (IQL and {SARSA or CRL}).** To see whether the value or the policy imposes a bigger bottleneck, we measure performance with varying amounts of data for the value and the policy. The color gradients (⇑, ⬈, ⇒) of these matrices reveal how the performance of offline RL is bottlenecked in each setting.

Table 1: **DDPG+BC is often the best policy extraction method.** We aggregate the performances over the entire data-scaling matrix and then over 8 random seeds in each setting. Scores at or above 95% of the best score are highlighted in bold. The table shows that DDPG+BC is better than or as good as AWR in **15 out of 16 settings**. We note that policy extraction hyperparameters are individually tuned for each setting (Figure 11).

| Task (Value Algorithm) | AWR | DDPG+BC | SfBC | | Task (Value Algorithm) | AWR | DDPG+BC | SfBC |
|---|---|---|---|---|---|---|---|---|
| gc-antmaze-large (IQL) | $51_{\pm 2}$ | $\mathbf{58}_{\pm 2}$ | $\mathbf{58}_{\pm 1}$ | | exorl-cheetah (IQL) | $71_{\pm 1}$ | $\mathbf{101}_{\pm 2}$ | $77_{\pm 2}$ |
| gc-antmaze-large (CRL) | $37_{\pm 2}$ | $\mathbf{58}_{\pm 2}$ | $51_{\pm 2}$ | | exorl-cheetah (SARSA) | $78_{\pm 1}$ | $\mathbf{131}_{\pm 3}$ | $89_{\pm 1}$ |
| antmaze-large (IQL) | $12_{\pm 2}$ | $17_{\pm 4}$ | $\mathbf{24}_{\pm 3}$ | | d4rl-hopper (IQL) | $\mathbf{53}_{\pm 1}$ | $\mathbf{52}_{\pm 1}$ | $43_{\pm 1}$ |
| antmaze-large (SARSA) | $\mathbf{0}_{\pm 0}$ | $\mathbf{0}_{\pm 0}$ | $\mathbf{0}_{\pm 0}$ | | d4rl-hopper (SARSA) | $56_{\pm 1}$ | $\mathbf{61}_{\pm 3}$ | $50_{\pm 2}$ |
| kitchen (IQL) | $80_{\pm 1}$ | $\mathbf{86}_{\pm 1}$ | $75_{\pm 1}$ | | d4rl-walker2d (IQL) | $73_{\pm 1}$ | $\mathbf{81}_{\pm 1}$ | $68_{\pm 1}$ |
| kitchen (SARSA) | $79_{\pm 1}$ | $\mathbf{83}_{\pm 1}$ | $73_{\pm 1}$ | | d4rl-walker2d (SARSA) | $79_{\pm 1}$ | $\mathbf{84}_{\pm 0}$ | $\mathbf{81}_{\pm 1}$ |
| exorl-walker (IQL) | $99_{\pm 1}$ | $\mathbf{191}_{\pm 6}$ | $140_{\pm 1}$ | | gc-roboverse (IQL) | $\mathbf{23}_{\pm 2}$ | $20_{\pm 2}$ | $14_{\pm 2}$ |
| exorl-walker (SARSA) | $94_{\pm 0}$ | $\mathbf{193}_{\pm 5}$ | $125_{\pm 1}$ | | gc-roboverse (CRL) | $13_{\pm 1}$ | $\mathbf{16}_{\pm 2}$ | $15_{\pm 1}$ |

deviations). In each matrix, we individually tune the hyperparameter for policy extraction ($\alpha$ or $N$) for each entry. These matrices show how performance varies with different amounts of data for the value and the policy. In our analysis, we specifically focus on the *color gradients* of these matrices, which reveal the main limiting factor behind the performance of offline RL in each setting. Note that the color gradients are mostly either vertical, horizontal, or diagonal. Vertical (⇑) color gradients indicate that the performance is most strongly affected by the amount of *policy* data, horizontal (⇒) gradients indicate it is mostly affected by *value* data, and diagonal (⬈) gradients indicate both.

Side-by-side comparisons of the data-scaling matrices from different policy extraction methods in Figure 1 suggest that, perhaps surprisingly, **different policy extraction algorithms often lead to significantly different performance and data-scaling behaviors, even though they extract policies from the *same* value function** (recall that the use of decoupled algorithms allows us to train a single value function, but use it for policy extraction in different ways). For example, on exorl-walker and exorl-cheetah, AWR performs remarkably poorly compared to DDPG+BC or SfBC on both value learning algorithms. Such a performance gap between policy extraction algorithms exists even when the value function is far from perfect, as can be seen in the low-data regimes in gc-antmaze-large and kitchen. In general, we find that the choice of a policy extraction procedure affects performance

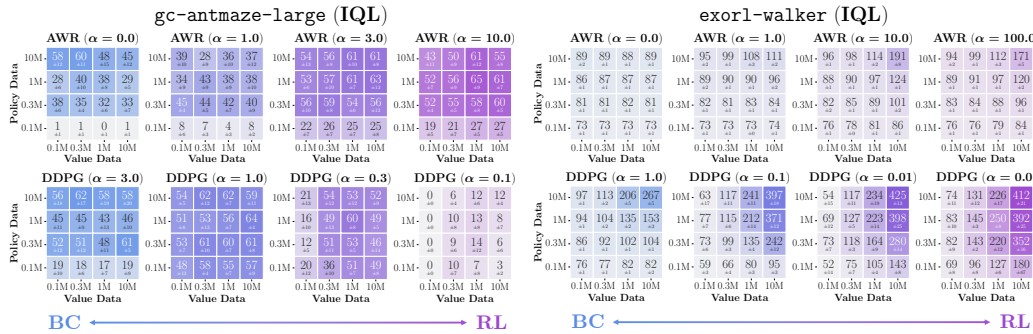

Figure 2: **Data-scaling matrices of AWR and DDPG+BC with different BC strengths ($\alpha$).** In `gc-antmaze-large`, AWR is *always* policy-bounded (⬆), but DDPG+BC has *both* policy-bounded (⬆) and value-bounded (➡) modes, depending on the value of $\alpha$. Notably, an in-between value of $\alpha = 1.0$ in DDPG+BC leads to the best of both worlds (see the bottom left corner of `gc-antmaze-large` with 0.1M datasets)!

often more than the choice of a value learning objective except `antmaze-large`, where the value function must be learned from sparse-reward, suboptimal datasets with long-horizon trajectories.

Among policy extraction algorithms, we find that **DDPG+BC almost always achieves the best performance and scaling behaviors across the board**, followed by SfBC, and the performance of AWR falls significantly behind the other two extraction algorithms in many cases (Table 1). Notably, the data-scaling matrices of AWR always have vertical (⬆) or diagonal (⬈) color gradients, implying that it does not fully utilize the value function (see Section 4.3 for clearer evidence). In other words, a non-careful choice of the policy extraction algorithm (*e.g.*, weighted behavioral cloning) hinders the use of learned value functions, imposing an unnecessary bottleneck on the performance of offline RL.

### 4.3 Deep dive 1: How different are the scaling properties of AWR and DDPG+BC?

To gain further insights into the difference between value-weighted behavioral cloning (*e.g.*, AWR) and behavior-regularized policy gradient (*e.g.*, DDPG+BC), we draw data-scaling matrices with different values of $\alpha$ (in Equations (1) and (2)), a hyperparameter that interpolates between RL and BC. Note that $\alpha = 0$ corresponds to BC in AWR and $\alpha = \infty$ corresponds to BC in DDPG+BC. We recall that the previous results (Figure 1) use the best temperature for each matrix entry (*i.e.*, aggregated by the maximum over temperatures), but here we show the full results with individual hyperparameters.

Figure 2 highlights the results on `gc-antmaze-large` and `exorl-walker` (see Appendix E for the full results). The results on `gc-antmaze-large` show a clear difference in scaling matrices between AWR and DDPG+BC. That is, AWR is *always* policy-bounded regardless of the BC strength $\alpha$ (*i.e.*, vertical (⬆) color gradients), whereas DDPG+BC has two "modes": it is policy-bounded (⬆) when $\alpha$ is large, and value-bounded (➡) and when $\alpha$ is small. Intriguingly, an in-between value of $\alpha = 1.0$ in DDPG+BC enables having the best of both worlds, significantly boosting performances across the entire matrix (note that it achieves very strong performance even with a 0.1M-sized dataset)! This difference in scaling behaviors suggests that the use of the learned value function in weighted behavioral cloning is limited. This becomes more evident in `exorl-walker` (Figure 2), where AWR fails to achieve strong performance even with a very high temperature value ($\alpha = 100$).

### 4.4 Deep dive 2: *Why* is DDPG+BC better than AWR?

We have so far seen several empirical results that suggest behavior-regularized policy gradient (*e.g.*, DDPG+BC) should be preferred to weighted behavioral cloning (*e.g.*, AWR) in any case. What makes DDPG+BC so much better than AWR? There are three potential reasons.

First, AWR only has a *mode-covering* weighted behavioral cloning term, while DDPG+BC has both *mode-seeking* first-order value maximization and *mode-covering* behavioral cloning terms. As a result, actions learned by AWR always lie within the convex hull of dataset actions, whereas DDPG+BC can "hillclimb" the learned value function, even allowing extrapolation to some degree while not deviating too far away from the mode. This not only enables a better use of the value function but produces a wider range of actions. To illustrate this, we plot test-time action

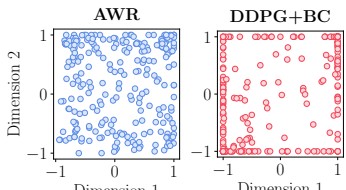

Figure 3: **AWR vs. DDPG actions.**

sampled from policies learned by AWR and DDPG+BC on `exorl-walker`. Figure 3 shows that AWR actions are relatively centered around the origin, while DDPG+BC actions are more spread out, which can sometimes help achieve an even higher degree of optimality.

Second, value-weighted behavioral cloning uses a much smaller number of *effective* samples than behavior-regularized policy gradient methods, especially when the temperature ($\alpha$) is large. This is because a small number of high-advantage transitions can potentially dominate learning signals for AWR (*e.g.*, a single transition with a weight of $e^{10}$ can dominate other transitions with smaller weights like $e^2$). As a result, AWR effectively uses only a fraction of datapoints for policy learning, being susceptible to overfitting. On the other hand, DDPG+BC is based on first-order maximization of the value function without any weighting, and thus is free from such an issue. Figure 4 illustrates this, where we compare the training and validation policy losses of AWR and DDPG+BC on `gc-antmaze-large` with the smallest 0.1M

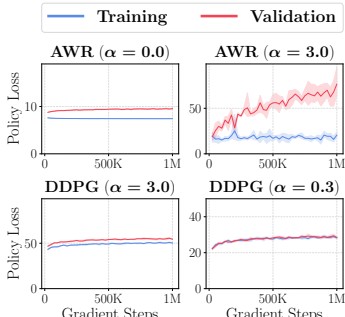

Figure 4: **AWR overfits.**

dataset (8 seeds). The results show that AWR with a large temperature ($\alpha = 3.0$) causes severe overfitting. Indeed, Figure 1 shows DDPG+BC often achieves significantly better performance than AWR in low-data regimes.

Third, AWR has a theoretical pathology in the regime with limited samples: since the coefficient multiplying $\log \pi(a \mid s)$ in the AWR objective (Equation (1)) is always positive, AWR can increase the likelihood of *all* dataset actions, regardless of how optimal they are. If the training dataset covers all possible actions, then the condition for normalization of the probability density function of $\pi(a \mid s)$ would alleviate this issue, but this coverage assumption is rarely achieved in practice. Under limited data coverage, and especially when the policy network is highly expressive and dataset states are unique (*e.g.*, continuous control problems), AWR can in theory *memorize* all state-action pairs in the dataset, potentially reverting to *unweighted* behavioral cloning.

> **Takeaway: Current policy extraction can inhibit effective use of the value function.**
>
> Do *not* use value-weighted behavior cloning (*e.g.*, AWR); use behavior-constrained policy gradient (*e.g.*, DDPG+BC), regardless of the value learning objective. This enables better scaling of performance with more data and better use of the value function.

## 5 Empirical analysis 2: Policy generalization (B3)

We now turn our focus to the third hypothesis, that the degree to which the agent **generalizes** to states that it visits at the evaluation time has a significant impact on performance. This is a unique bottleneck to the *offline* RL problem setting, where the agent encounters new, potentially out-of-distribution states at test time.

### 5.1 Analysis setup

To understand this bottleneck concretely, we first define three key metrics quantifying a notion of *accuracy* of a given policy in terms of distances against the optimal policy. Specifically, we use the following mean squared error (MSE) metrics to quantify policy accuracy:

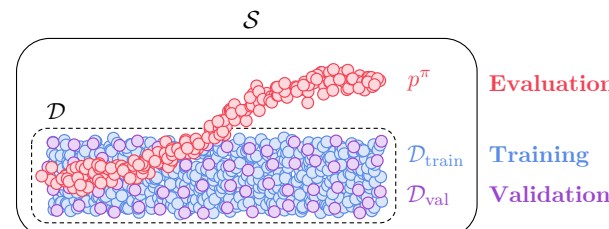

Figure 5: **Three distributions for the MSE metrics.**

$$\text{(Training MSE)} = \mathbb{E}_{s \sim \mathcal{D}_{\text{train}}}[(\pi(s) - \pi^*(s))^2], \tag{4}$$

$$\text{(Validation MSE)} = \mathbb{E}_{s \sim \mathcal{D}_{\text{val}}}[(\pi(s) - \pi^*(s))^2], \tag{5}$$

$$\text{(Evaluation MSE)} = \mathbb{E}_{s \sim p^\pi(\cdot)}[(\pi(s) - \pi^*(s))^2], \tag{6}$$

where $\mathcal{D}_{\text{train}}$ and $\mathcal{D}_{\text{val}}$ respectively denote the training and validation datasets, $\pi^*$ denotes an optimal policy, which we assume access to for evaluation and visualization purposes only. Validation MSE

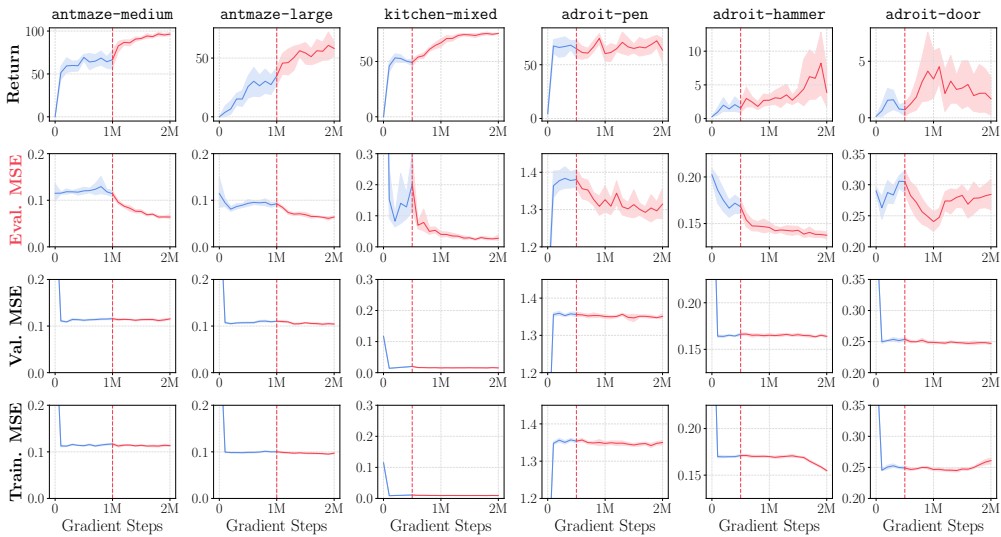

Figure 6: **How do offline RL policies improve with additional interaction data?** In many environments, offline-to-online RL *only* improves evaluation MSEs, while validation MSEs and training MSEs often *remain completely flat* (see Section 5 for the definitions of these metrics). This suggests that current offline RL algorithms may already be great at learning an effective policy on *in-distribution* states, and the performance of offline RL is often mainly determined by how well the policy *generalizes* on its own state distribution at test time.

measures the policy accuracy on states sampled from the *same* dataset distribution as the training distribution (*i.e.*, in-distribution MSE, Figure 5), while evaluation MSE measures the policy accuracy on states the agent visits at test time, which can potentially be very different from the dataset distribution (*i.e.*, out-of-distribution MSE, Figure 5). We note that, while these metrics might not always be perfectly indicative of the performance of a policy (see Appendix A), they serve as convenient proxies to estimate policy accuracy in many continuous-control domains in practice.

One way to measure the degree to which test-time generalization affects performance is to evaluate how much room there is for various policy MSE metrics to improve when further training on additional policy rollouts is allowed. The distribution of states induced by rolling out the policy is an ideal distribution to improve performance, as the policy receives direct feedback on its own actions at the states it would visit. Hence, by tracking the extent to which various MSEs improve and how their predictive power towards performance evolves over online interaction, we will be able to understand which is a bigger bottleneck: in-distribution generalization (*i.e.*, improvements towards validation MSE under the offline dataset distribution) or out-of-distribution generalization (*i.e.*, improvements in evaluation MSE under the on-policy state distribution). To this end, we measure these three types of MSEs over the course of online interaction, when learning from a policy trained on offline data only (*i.e.*, the *offline-to-online* RL setting). Specifically, we train offline-to-online IQL agents on six D4RL [12] tasks (`antmaze-{medium, large}`, `kitchen`, and `adroit-{pen, hammer, door}`), and measure the MSEs with pre-trained expert policies that approximate $\pi^*$ (see Appendix D.4).

## 5.2 Results: Test-time generalization is often the main bottleneck in offline RL

Figure 6 shows the results (8 seeds with 95% confidence intervals), where we denote online training steps in red. The results show that, perhaps surprisingly, in many environments continued training with online interaction *only* improves evaluation MSEs, while training and validation MSEs often *remain completely flat* during online training. Also, we can see that the evaluation MSE is the most predictive of the performance of offline RL among the three metrics. In other words, the results show that, despite the fact that on-policy data provides for an oracle distribution to improve policy accuracy, performance improvement is often only reflected in the evaluation MSEs computed under the policy's own state distribution.

What does this tell us? This indicates that, current offline RL methods may already be sufficiently great at learning the best possible policy *within the distribution of states covered by the offline dataset*, and **the agent's performance is often mainly determined by how well it *generalizes* under its own state distribution at test time**, as suggested by the fact that evaluation MSE is most predictive of performance. This finding somewhat contradicts prior beliefs: while algorithmic techniques in offline RL largely attempt to improve policy optimality on *in-distribution states* (by addressing the issue with out-of-distribution *actions*), our results suggest that modern offline RL algorithms may

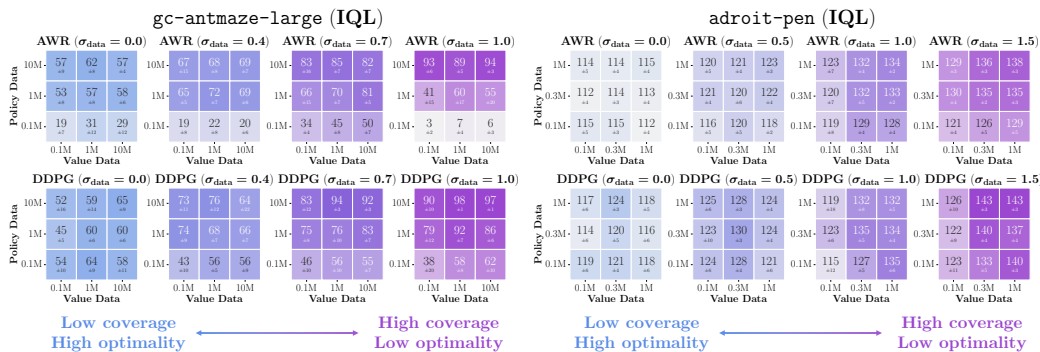

Figure 7: **Should we use high-coverage or high-optimality datasets?** The data-scaling matrices above show that *high-coverage* datasets can be much more effective than high-optimality datasets. This is because high-coverage datasets can improve *test-time policy accuracy*, one of the main bottlenecks of offline RL.

already saturate on this axis. Further performance differences may simply be due to the effects of a given offline RL objective on *novel states*, which very few methods explicitly control!

That said, controlling test-time generalization might also appear impossible: while offline RL methods could hillclimb on validation accuracy via a combination of techniques that address statistical errors such as regularization (*e.g.*, Dropout [52], LayerNorm [3], etc.), improving *test-time* policy accuracy requires generalization to a potentially very different *distribution* (Figure 5), which is theoretically impossible to guarantee without additional coverage or structural assumptions, as the test-time state distribution can be arbitrarily adversarial in the worst case. However, we claim that if we actively utilize the information available at test time or have the freedom to design offline datasets, it is possible to improve test-time policy accuracy in practice, and we discuss such solutions below (see Appendix C for further discussions).

### 5.3 Solution 1: Improve offline data coverage

If we have the freedom to control the data collection process, perhaps the most straightforward way to improve test-time policy accuracy is to use a dataset that has as *high coverage* as possible so that test-time states can be covered by the dataset distribution. However, at the same time, high-coverage datasets often involve exploratory actions, which may compromise the quality (optimality) of the dataset. This makes us wonder in practice: *which is more important, high coverage or high optimality?*

To answer this question, we revert back to our analysis tool of data-scaling matrices from Section 4 and empirically compare the data-scaling matrices on datasets collected by expert policies with different levels of action noises ($\sigma_{\text{data}}$). Figure 7 shows the results of IQL agents on `gc-antmaze-large` and `adroit-pen` (8 seeds each). The results suggest that the performance of offline RL generally improves as the dataset has better state coverage, despite the increase in suboptimality. This is aligned with our findings in Figure 6, which indicate that the main challenge of offline RL is often *not* learning an effective policy from suboptimal data, but rather learning a policy that generalizes well to test-time states. In addition, we note that it is crucial to use a value gradient-based policy extraction method (DDPG+BC; see Section 4) in this case as well, where we train a policy from high-coverage data. For instance, in low-data regimes in `gc-antmaze-large` in Figure 7, AWR fails to fully leverage the value function, whereas DDPG+BC still allows the algorithm to improve performance with better value functions. Based on our findings, we suggest practitioners prioritize *high coverage* (particularly around the states that the optimal policy will likely visit) over high optimally when collecting datasets.

### 5.4 Solution 2: Test-time policy improvement

If we do not wish to modify offline data collection, another way to improve test-time policy accuracy is to *on-the-fly* train or steer the policy guided by the learned value function on *test-time states*. Especially given that imperfect policy extraction from the value function is often a significant bottleneck in offline RL (Section 4), we propose two simple techniques to further distill the information in the value function into the policy on test-time states.

**(1) On-the-fly policy extraction (OPEX).** Our first idea is to simply adjust policy actions in the direction of the value gradient at evaluation time. Specifically, after sampling an action from the policy $a \sim \pi(\cdot \mid s)$ at test time, we further adjust the action based on the *frozen* learned $Q$ function during evaluation rollouts with the following formula:

$$a \leftarrow a + \beta \cdot \nabla_a Q(s, a), \tag{7}$$

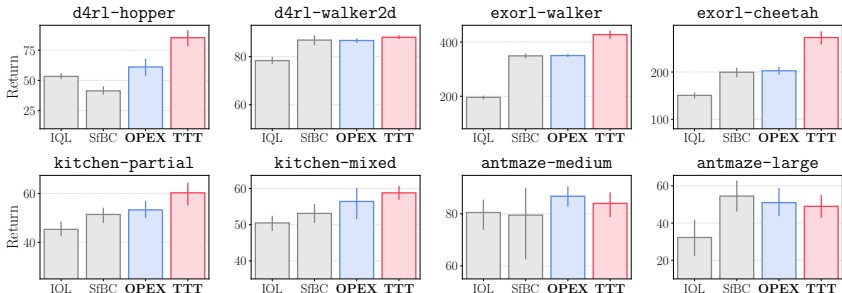

Figure 8: **Test-time policy improvement strategies (OPEX and TTT).** Our two on-the-fly policy improvement techniques (OPEX and TTT) lead to substantial performance improvements on diverse tasks, by mitigating the test-time policy generalization bottleneck.

where $\beta$ is a hyperparameter that corresponds to the test-time "learning rate". Intuitively, Equation (7) adjusts the action in the direction that maximally increases the learned Q function. We call this technique **on-the-fly policy extraction (OPEX)**. Note that OPEX requires only *a single line of additional code* at evaluation and does not change the training procedure at all.

**(2) Test-time training (TTT).** We also propose another variant that further updates the parameters of the policy by continuously extracting the policy from the fixed value function on test-time states, as more rollouts are performed. Specifically, we update the policy $\pi$ with the following objective:

$$\max_{\pi} \; \mathcal{J}_{\text{TTT}}(\pi) = \mathbb{E}_{s,a\sim\mathcal{D}\,\cup\,p^{\pi}(\cdot)}[Q(s,\mu^{\pi}(s)) - \beta \cdot D_{\text{KL}}(\pi^{\text{off}} \parallel \pi)], \tag{8}$$

where $\pi^{\text{off}}$ denotes the fixed, learned offline RL policy, $\mathcal{D}\cup p^{\pi}(\cdot)$ denotes the mixture of the dataset and evaluation state distributions, and $\beta$ denotes a hyperparameter that controls the strength of the regularizer. Intuitively, Equation (8) is a "parameter-updating" version of OPEX, where we further update the parameters of the policy $\pi$ to maximize the learned value function, while not deviating too far away from the learned offline RL policy. We call this scheme **test-time training (TTT)**. Note that TTT only trains $\pi$ based on test-time interaction data, while $Q$ and $\pi^{\text{off}}$ remain fixed.

Figure 8 compares the performances of vanilla IQL, SfBC (Equation (3), another test-time policy extraction method that does not involve gradients), and our two gradient-based test-time policy improvement strategies on eight tasks (8 seeds each, error bars denote 95% confidence intervals). The results show that OPEX and TTT improve performance over vanilla IQL and SfBC in many tasks, often by significant margins, by mitigating the test-time policy generalization bottleneck.

> **Takeaway: Improving test-time policy accuracy significantly boosts performance.**
>
> Test-time policy *generalization* is one of the most significant bottlenecks in offline RL. Use high-coverage datasets. Improve policy accuracy on test-time states with on-the-fly policy improvement techniques.

## 6 Conclusion: What does our analysis tell us?

In this work, we empirically demonstrated that, contrary to the prior belief that improving the quality of the value function is the primary bottleneck of offline RL, current offline RL methods are often heavily limited by how faithfully the policy is *extracted* from the value function and how well this policy *generalizes* on test-time states. **For practitioners**, our analysis suggests a clear empirical recipe for effective offline RL: train a value function on as *diverse* data as possible, and allow the policy to maximally utilize the value function, with the best policy extraction objective (*e.g.*, DDPG+BC) and/or potential test-time policy improvement strategies. **For future algorithms research**, our analysis emphasizes two important open questions in offline RL: (1) What is the best way to *extract* a policy from the learned value function? (2) How can we train a policy in a way that it *generalizes* well on test-time states? The second question is particularly notable, because it suggests a diametrically opposed viewpoint to the prevailing theme of pessimism in offline RL, where only a few works have explicitly aimed to address this generalization aspect of offline RL [37, 38, 63]. We believe finding effective answers to these questions would lead to significant performance gains in offline RL, substantially enhancing its applicability and scalability, and would encourage the community to incorporate a holistic picture of offline RL alongside the current prominent research on value function learning.

## Acknowledgments

We thank Benjamin Eysenbach and Dibya Ghosh for insightful discussions about data-scaling matrices and state representations, respectively, and Oleh Rybkin, Fahim Tajwar, Mitsuhiko Nakamoto, Yingjie Miao, Sandra Faust, and Dale Schuurmans for helpful feedback on earlier drafts of this work. This work was partly supported by the Korea Foundation for Advanced Studies (KFAS), National Science Foundation Graduate Research Fellowship Program under Grant No. DGE 2146752, and ONR N00014-21-1-2838. This research used the Savio computational cluster resource provided by the Berkeley Research Computing program at UC Berkeley.

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

# Appendices

## A   Limitations

One limitation of our analysis is that the MSE metrics in Equations (4) to (6) are in some sense "proxies" to measure the accuracy of the policy (somewhat similarly to how Bellman errors do not always accurately reflect value errors in the context of value learning [16]). For instance, if there exist multiple optimal actions that are potentially very different from one another, or the expert policy used in practice is not sufficiently optimal, the MSE metrics might not be highly indicative of the performance or accuracy of the policy. Nonetheless, we empirically find that there is a strong correlation between the evaluation MSE metric and performance, and we believe our analysis could further be refined with potentially more sophisticated metrics (*e.g.*, by considering $\mathbb{E}[Q^*(s, a)]$ instead of $\mathbb{E}[(\pi(s) - \pi^*(s))^2]$), which we leave for future work.

Another limitation of our analysis in Section 4 is we only consider policy extraction in continuous-action environments. In discrete-action environments, our takeaway might not directly apply in its current form because (1) DDPG+BC is not straightforwardly defined with discrete actions and (2) it is possible to directly use the Q function to implicitly define a policy (without having a separate policy network). We leave investigating the effect of policy extraction in discrete-action environments for future work.

## B   Preliminaries

We consider a Markov decision process (MDP) defined by $\mathcal{M} = (\mathcal{S}, \mathcal{A}, r, \mu, p)$. $\mathcal{S}$ denotes the state space, $\mathcal{A}$ denotes the action space, $r : \mathcal{S} \times \mathcal{A} \to \mathbb{R}$ denotes the reward function, $\mu \in \Delta(\mathcal{S})$ denotes the initial state distribution, and $p : \mathcal{S} \times \mathcal{A} \to \Delta(\mathcal{S})$ denotes the transition dynamics, where $\Delta(\mathcal{X})$ denotes the set of probability distributions over a set $\mathcal{X}$. We consider the offline RL problem, whose goal is to find a policy $\pi : \mathcal{S} \to \Delta(\mathcal{A})$ (or $\pi : \mathcal{S} \to \mathcal{A}$ if deterministic) that maximizes the discount return $J(\pi) = \mathbb{E}_{\tau \sim p^\pi(\tau)}[\sum_{t=0}^{T} \gamma^t r(s_t, a_t)]$, where $p^\pi(\tau) = p^\pi(s_0, a_0, s_1, a_1, \ldots, s_T, a_T) = \mu(s_0)\pi(a_0 \mid s_0)p(s_1 \mid s_0, a_0) \cdots \pi(a_T \mid s_T)$ and $\gamma$ is a discount factor, solely from a static dataset $\mathcal{D} = \{\tau_i\}_{i \in \{1, 2, \ldots, N\}}$ without online interactions. In some experiments, we consider offline *goal-conditioned* RL [2, 11, 22, 44, 57] as well, where the policy and reward function are also conditioned on a goal state $g$, which is sampled from a goal distribution $p_g \in \Delta\mathcal{S}$. For goal-conditioned RL, we assume a sparse goal-conditioned reward function, $r(s, g) = \mathbb{1}(s = g)$, which does not require any prior knowledge about the state space. We also assume that the episode ends upon goal-reaching [44, 45, 57].

## C   Policy generalization: Rethinking the role of state representations

In this section, we introduce another way to improve test-time policy accuracy from the perspective of *state representations*. Specifically, we claim that we can improve test-time policy accuracy by using a "good" representation that *naturally* enables out-of-distribution generalization. Since this might sound a bit cryptic, we first show results to illustrate this point.

Figure 9 shows the performances of goal-conditioned BC[1] on `gc-antmaze-large` with two different *homeomorphic* representations: one with the original state representation $s$, and one with a different representation $\phi(s)$ with a continuous, *invertible* $\phi$ (specifically, $\phi$ transforms $x$-$y$ coordinates with invertible $\tanh$ kernels; see Appendix D.6). Hence, these two representations contain the exactly same amount of information and are even topologically homeomorphic (under the standard Euclidean topology). However, they result in *very* different performances, and

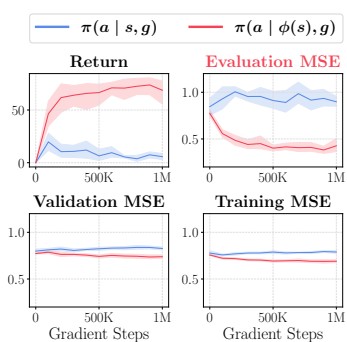

Figure 9: **A good state representation naturally enables test-time generalization, leading to substantially better performance.**

---

[1]Here, we use BC (not RL) to focus solely on state representations, obviating potential confounding factors regarding the value function.

the MSE plots in Figure 9 indicate that this difference is due to
nothing other than the better test-time, *evaluation* MSE (observe that their training and validation
MSEs are nearly identical)!

This result sheds light on an important perspective of state representations: a good state representation should be able to enable *test-time generalization naturally*. While designing such a good state representation might require some knowledge or inductive biases about the task, our results suggest that using such a representation is nonetheless very important in practice, since it affects the performance of offline RL significantly by improving test-time policy generalization capability.

## D   Experimental details

We provide the full experimental details in this section.

### D.1   Value learning objectives

**One-step RL (SARSA)**. SARSA [5] is one of the simplest offline value learning algorithms. Instead of fitting a Bellman optimal value function $Q^*$, SARSA aims to fit a behavioral value function $Q^\beta$ with TD-learning, without querying out-of-distribution actions. Concretely, SARSA minimizes the following loss:

$$\min_Q \mathcal{L}_{\text{SARSA}}(Q) = \mathbb{E}_{(s,a,s',a')\sim\mathcal{D}}[(r(s,a) + \gamma\bar{Q}(s',a') - Q(s,a))^2], \tag{9}$$

where $s'$ and $a'$ denote the next state and action, respectively, and $\bar{Q}$ denotes the target $Q$ network [39]. Despite its apparent simplicity, extracting a policy by maximizing the value function learned by SARSA is known to be a surprisingly strong baseline [5, 30].

**Implicit Q-learning (IQL)**. Implicit Q-learning (IQL) [25] aims to fit a Bellman optimal value function $Q^*$ by approximating the maximum operator with an in-sample expectile regression. IQL minimizes the following losses:

$$\min_Q \mathcal{L}_{\text{IQL}}^Q(Q) = \mathbb{E}_{(s,a,s')\sim\mathcal{D}}[(r(s,a) + \gamma V(s') - Q(s,a))^2], \tag{10}$$

$$\min_V \mathcal{L}_{\text{IQL}}^V(V) = \mathbb{E}_{(s,a)\sim\mathcal{D}}[\ell_\tau^2(\bar{Q}(s,a) - V(s))], \tag{11}$$

where $\ell_\tau^2(x) = |\tau - \mathbb{1}(x < 0)|x^2$ is the expectile loss [43] with an expectile parameter $\tau$. Intuitively, when $\tau > 0.5$, the expectile loss in Equation (11) penalizes positive errors more than negative errors, which makes $V$ closer to the maximum value of $\bar{Q}$. This way, IQL approximates $V^*$ and $Q^*$ only with in-distribution dataset actions, without referring to the erroneous values at out-of-distribution actions.

**Contrastive RL (CRL)**. Contrastive RL (CRL) [11] is a value learning algorithm for offline goal-conditioned RL based on contrastive learning. CRL maximizes the following objective:

$$\max_f \mathcal{J}_{\text{CRL}}(f) = \mathbb{E}_{s,a\sim\mathcal{D},g\sim p_\mathcal{D}^+(\cdot|s,a),g^-\sim p_\mathcal{D}^+(\cdot)}[\log\sigma(f(s,a,g)) + \log(1 - \sigma(f(s,a,g^-)))], \tag{12}$$

where $\sigma$ denotes the sigmoid function and $p_\mathcal{D}^+(\cdot \mid s,a)$ denotes the geometric future state distribution of the dataset $\mathcal{D}$. Eysenbach et al. [11] show that the optimal solution of Equation (12) is given as $f^*(s,a,g) = \log(p_\mathcal{D}^+(g \mid s,a)/p_\mathcal{D}^+(g))$, which gives us the behavioral goal-conditioned Q function as $Q^\beta(s,a,g) = p_\mathcal{D}^+(g \mid s,a) = p_\mathcal{D}^+(g)e^{f^*(s,a,g)}$, where $p_\mathcal{D}^+(g)$ is a policy-independent constant.

### D.2   Environments and datasets

We describe the environments and datasets we employ in our analysis.

#### D.2.1   Data-scaling analysis

For the data-scaling analysis in Section 4, we employ the following environments and datasets (Figure 10).

- `antmaze-large` and `gc-antmaze-large` are based on the `antmaze-large-diverse-v2` environment from the D4RL suite [12], where the agent must be able to manipulate a quadrupedal robot to reach a given target goal (`antmaze-large`) or to reach any goal from any other state

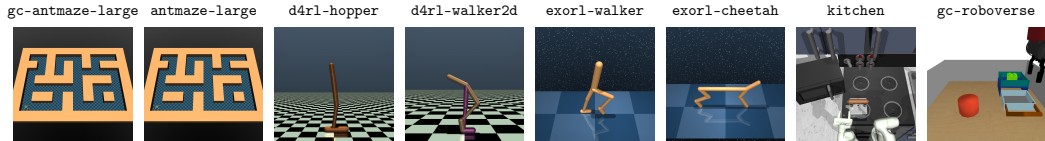

Figure 10: **Environments.**

(gc-antmaze-large) in a given maze. For the dataset for gc-antmaze-large in our data-scaling analysis, we collect 10M transitions using a noisy expert policy that navigates through the maze. We use the same policy and noise level ($\sigma_{\text{data}} = 0.2$) as the one used to collect antmaze-large-diverse-v2 in D4RL.

- d4rl-hopper and d4rl-walker2d are the hopper-medium-v2 and walker2d-medium-v2 tasks from the D4RL locomotion suite. We use the original 1M-sized datasets collected by partially trained policies [12].

- exorl-walker and exorl-cheetah are the walker-run and cheetah-run tasks from the ExORL benchmark [64]. We use the original 10M-sized datasets collected by RND agents [6]. Since the datasets are collected by purely unsupervised exploratory policies, they feature high suboptimality and high state-action diversity.

- kitchen is based on the kitchen-mixed-v0 task from the D4RL suite, where the goal is to complete four manipulation tasks (*e.g.*, opening the microwave, moving the kettle) with a robot arm. Since the original dataset size is relatively small, for our data-scaling analysis, we collect a large 1M-sized dataset with a noisy, biased expert policy, where we add noises sampled from a zero-mean Gaussian distribution with a standard deviation of 0.2 in addition to a randomly initialized policy's actions to the expert policy's actions.

- gc-roboverse is a pixel-based goal-conditioned robotic task, where the goal is to manipulate a robot arm to rearrange objects to match a target image. The agent must be able to perform object manipulation purely from $48 \times 48 \times 3$ images. We use the 1M-sized dataset used by Park et al. [44], Zheng et al. [65].

### D.2.2 Policy generalization analysis

For the policy generalization analysis in Section 5, we use the antmaze-medium-diverse-v2, antmaze-large-diverse-v2, kitchen-partial-v0, kitchen-mixed-v0, pen-cloned-v1, hammer-cloned-v1, door-cloned-v1, hopper-medium-v2, and walker2d-medium-v2 environments and datasets from the D4RL suite [12] as well as the walker-run and cheetah-run from the ExORL suite [64].

### D.3 Data-scaling matrices

We train agents for 1M steps (500K steps for gc-roboverse) with each pair of value learning and policy extraction algorithms. We evaluate the performance of the agent every 100K steps with 50 rollouts, and report the performance averaged over the last 3 evaluations and over 8 seeds. In Figures 1 and 7, we individually tune the policy extraction hyperparameter ($\alpha$ for AWR and DDPG+BC, and $N$ for SfBC) for each cell, and report the performance with the best hyperparameter. To save computation, we extract multiple policies with different hyperparameters from the same value function (note that this is possible because we use decoupled offline RL algorithms). To generate smaller-sized datasets from the original full dataset, we randomly shuffle trajectories in the original dataset using a fixed random seed, and take the first $K$ trajectories such that smaller datasets are fully contained in larger datasets.

### D.4 MSE metrics

We randomly split the trajectories in a dataset into a training set (95%) and a validation set (5%) in our experiments. For the expert policies $\pi^*$ in the MSE metrics defined in Equations (4) to (6), we use either the original expert policies from the D4RL suite (adroit-{pen, hammer, door} and gc-antmaze-large) or policies pre-trained with offline-to-online RL until their performance saturates (antmaze-{medium, large} and kitchen-mixed). To train "global" expert policies for

`antmaze-{medium, large}`, we reset the agent to arbitrary locations in the entire maze. This initial state distribution is only used to train an expert policy; we use the original initial state distribution for the other experiments.

## D.5 Test-time policy improvement methods

In Figure 8, for IQL, SfBC, and OPEX, we train IQL agents (with original AWR) for 500K (`kitchen`) or 1M (others) gradient steps. For TTT, we further train the policy up to 2M gradient steps with a learning rate of $0.00003$. In `antmaze`, we consider both deterministic evaluation and stochastic evaluation with a fixed standard deviation of $0.4$ (which roughly matches the learned standard deviation of the BC policy), and report the best performance of them for each method.

## D.6 State representation experiments

We describe the state representation $\phi$ used in Appendix C. An `antmaze` state consists of a 2-D $x$-$y$ coordinates and 27-D proprioceptive information. We transform $x$ and $y$ individually with 32 $\tanh$ kernels, *i.e.*,

$$\tilde{x}_i = \tanh\left(\frac{x - x_i}{\delta_x}\right) \tag{13}$$

$$\tilde{y}_i = \tanh\left(\frac{y - y_i}{\delta_x}\right), \tag{14}$$

where $i \in \{1, 2, \ldots, 32\}$, $\delta_x = x_2 - x_1$, $\delta_y = y_2 - y_1$, and $x_1, \ldots, x_{32}$ and $y_1, \ldots, y_{32}$ are defined as `numpy.linspace(-2, 38, 32)` and `numpy.linspace(-2, 26, 32)`, respectively. Denoting the 27-D proprioceptive state as $s_{\text{proprio}}$, $\phi(s)$ is defined as follows: $\phi([x, y; s_{\text{proprio}}]) = [\tilde{x}_1, \ldots, \tilde{x}_{32}, \tilde{y}_1, \ldots, \tilde{y}_{32}; s_{\text{proprio}}]$, where ';' denotes concatenation. Intuitively, $\phi$ is similar to the discretization of the $x$-$y$ dimensions with 32 bins, but with a continuous, invertible $\tanh$ transformation instead of binary discretization.

## D.7 Implementation details

Our implementation is based on `jaxrl_minimal` [20] and the official implementation of HIQL [44] (for offline goal-conditioned RL). We use an internal cluster consisting of A5000 GPUs to run our experiments. Each experiment in our work takes no more than 18 hours.

### D.7.1 Data-scaling analysis

**Default hyperparameters.** We mostly follow the original hyperparameters for IQL [25], goal-conditioned IQL [44], and CRL [11]. Tables 2 and 3 list the common and environment-specific hyperparameters, respectively. For SARSA, we use the same implementation as IQL, but with the standard $\ell^2$ loss instead of an expectile loss. For pixel-based environments (*i.e.*, `gc-roboverse`), we use the same architecture and image augmentation as Park et al. [44]. In goal-conditioned environments as well as `antmaze` tasks, we subtract 1 from rewards, following previous works [25, 44].

**Policy extraction methods.** We use Gaussian distributions (without $\tanh$ squashing) to model action distributions. We use a fixed standard deviation of 1 for AWR and DDPG+BC and a learnable standard deviation for SfBC. For DDPG+BC, we clip actions to be within the range of $[-1, 1]$ in the deterministic policy gradient term in Equation (2). We empirically find that this is better than $\tanh$ squashing [14] across the board, and is important to achieving strong performance in some environments. We list the policy extraction hyperparameters we consider in our experiments in curly brackets in Table 3.

### D.7.2 Policy generalization analysis

**Hyperparameters.** Table 4 lists the hyperparameters that we use in our offline-to-online RL and test-time policy improvement experiments. In these experiments, we use Gaussian distributions with learnable standard deviations for action distributions.

Table 2: **Common hyperparameters for data-scaling matrices.**

| Hyperparameter | Value |
|---|---|
| Learning rate | 0.0003 |
| Optimizer | Adam [24] |
| Target smoothing coefficient | 0.005 |
| Discount factor $\gamma$ | 0.99 |

Table 3: **Environment-specific hyperparameters for data-scaling matrices.**

| Environment | gc-antmaze-large | antmaze-large | d4rl-hopper | d4rl-walker |
|---|---|---|---|---|
| # gradient steps | $10^6$ | $10^6$ | $10^6$ | $10^6$ |
| Minibatch size | 1024 | 256 | 256 | 256 |
| MLP dimensions | $(512, 512, 512)$ | $(256, 256)$ | $(256, 256)$ | $(256, 256)$ |
| IQL expectile | 0.9 | 0.9 | 0.7 | 0.7 |
| LayerNorm [3] | True | False | True | True |
| AWR $\alpha$ (IQL) | $\{0, 1, 3, 10\}$ | $\{0, 3, 10, 30\}$ | $\{0, 1, 3, 10\}$ | $\{0, 1, 3, 10\}$ |
| AWR $\alpha$ (SARSA/CRL) | $\{0, 10, 30, 100\}$ | $\{0, 3, 10, 30\}$ | $\{0, 1, 3, 10\}$ | $\{0, 1, 3, 10\}$ |
| DDPG+BC $\alpha$ (IQL) | $\{0.1, 0.3, 1, 3\}$ | $\{0.1, 0.3, 1, 3\}$ | $\{1, 3, 10, 30\}$ | $\{1, 3, 10, 30\}$ |
| DDPG+BC $\alpha$ (SARSA/CRL) | $\{0.1, 0.3, 1, 3\}$ | $\{0.1, 0.3, 1, 3\}$ | $\{1, 3, 10, 30\}$ | $\{1, 3, 10, 30\}$ |
| SfBC $N$ (IQL) | $\{1, 16, 64\}$ | $\{1, 16, 64\}$ | $\{1, 16, 64\}$ | $\{1, 16, 64\}$ |
| SfBC $N$ (SARSA/CRL) | $\{1, 16, 64\}$ | $\{1, 16, 64\}$ | $\{1, 16, 64\}$ | $\{1, 16, 64\}$ |

| Environment | exorl-walker | exorl-cheetah | kitchen | gc-roboverse |
|---|---|---|---|---|
| # gradient steps | $10^6$ | $10^6$ | $10^6$ | $5 \times 10^5$ |
| Minibatch size | 1024 | 1024 | 1024 | 256 |
| MLP dimensions | $(512, 512, 512)$ | $(512, 512, 512)$ | $(512, 512, 512)$ | $(512, 512, 512)$ |
| IQL expectile | 0.9 | 0.9 | 0.7 | 0.7 |
| LayerNorm [3] | True | True | False | True |
| AWR $\alpha$ (IQL) | $\{0, 1, 10, 100\}$ | $\{0, 1, 10, 100\}$ | $\{0, 1, 3, 10\}$ | $\{0, 0.1, 1, 10\}$ |
| AWR $\alpha$ (SARSA/CRL) | $\{0, 1, 10, 100\}$ | $\{0, 1, 10, 100\}$ | $\{0, 1, 3, 10\}$ | $\{0, 1, 10, 100\}$ |
| DDPG+BC $\alpha$ (IQL) | $\{0, 0.01, 0.1, 1\}$ | $\{0, 0.01, 0.1, 1\}$ | $\{10, 30, 100, 300\}$ | $\{3, 10, 30, 100\}$ |
| DDPG+BC $\alpha$ (SARSA/CRL) | $\{0, 0.01, 0.1, 1\}$ | $\{0, 0.01, 0.1, 1\}$ | $\{10, 30, 100, 300\}$ | $\{3, 10, 30, 100\}$ |
| SfBC $N$ (IQL) | $\{1, 16, 64\}$ | $\{1, 16, 64\}$ | $\{1, 16, 64\}$ | $\{1, 16, 64\}$ |
| SfBC $N$ (SARSA/CRL) | $\{1, 16, 64\}$ | $\{1, 16, 64\}$ | $\{1, 16, 64\}$ | $\{1, 16, 64\}$ |

# E   Additional results

We provide the full data-scaling matrices with different policy extraction hyperparameters ($\alpha$ for AWR and DDPG+BC, and $N$ for SfBC) in Figure 11.

Table 4: **Hyperparameters for policy generalization analysis.**

| Hyperparameter | Value |
|---|---|
| Learning rate | 0.0003 |
| Optimizer | Adam [24] |
| # offline gradient steps | $10^6$ (`antmaze`), $5 \times 10^5$ (`kitchen`, `adroit`) |
| # total gradient steps | $2 \times 10^6$ |
| # gradient steps per environment step | 1 |
| Minibatch size | 1024 (`kitchen`), 256 (`antmaze`, `adroit`) |
| MLP dimensions | $(512, 512, 512)$ (`kitchen`), $(256, 256)$ (`antmaze`, `adroit`) |
| Target smoothing coefficient | 0.005 |
| Discount factor $\gamma$ | 0.99 |
| LayerNorm [3] | True (`kitchen`), False (`antmaze`, `adroit`) |
| IQL expectile | 0.9 (`antmaze`), 0.7 (`kitchen`, `adroit`) |
| Policy extraction method | AWR |
| AWR $\alpha$ | 10 (`antmaze`), 0.5 (`kitchen`), 3 (`adroit`) |
| SfBC $N$ | 16 |
| OPEX $\beta$ | 0.3 (`antmaze`), 0.0003 (`kitchen`), 0.03 (`d4rl-hopper`), 0.1 (`d4rl-walker2d`), 1 (`exorl-{walker, cheetah}`) |
| TTT $\beta$ | 0.3 (`antmaze`), 5 (`kitchen`), 0.5 (`d4rl-hopper`), 0.3 (`d4rl-walker2d`), 0.01 (`exorl-{walker, cheetah}`) |

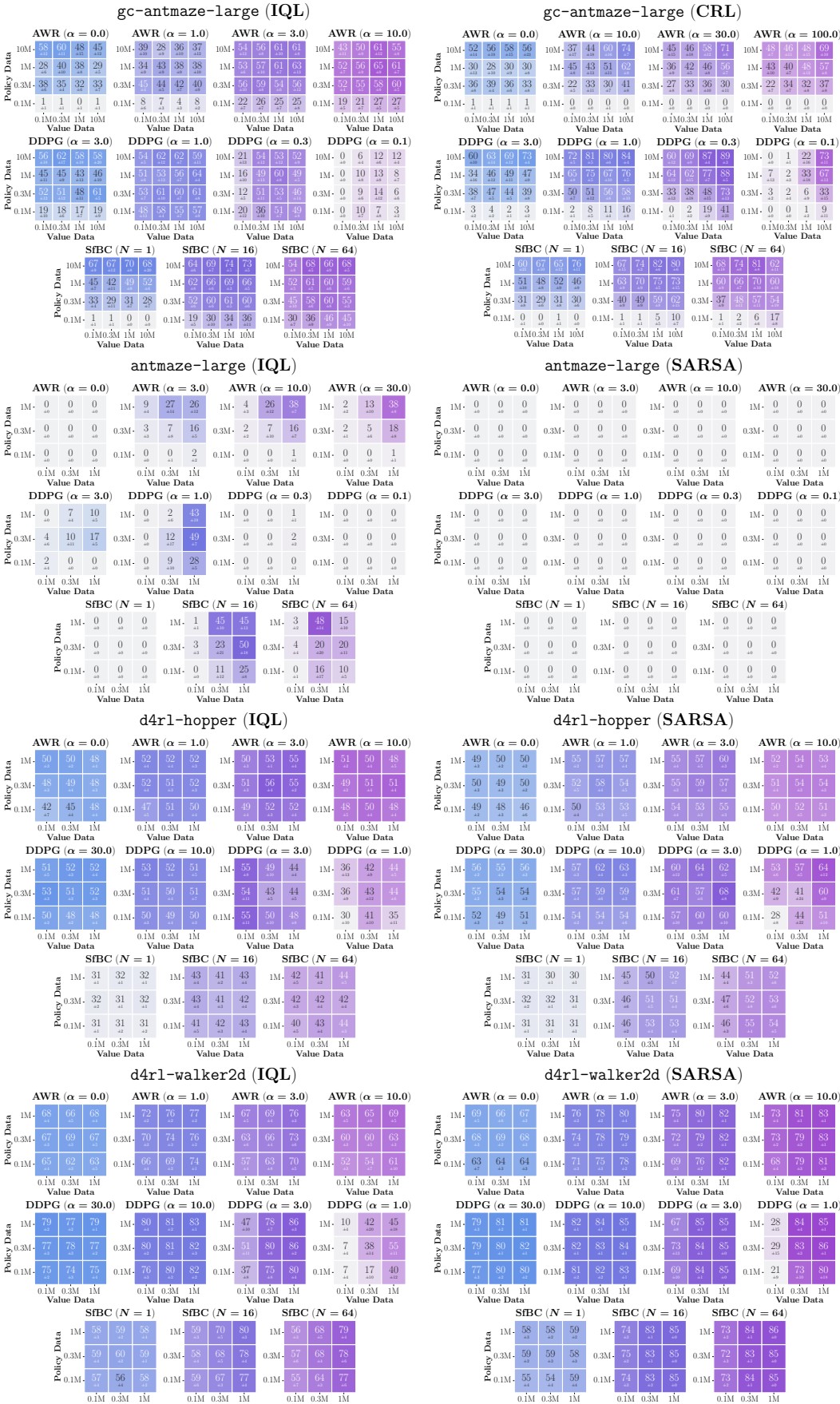

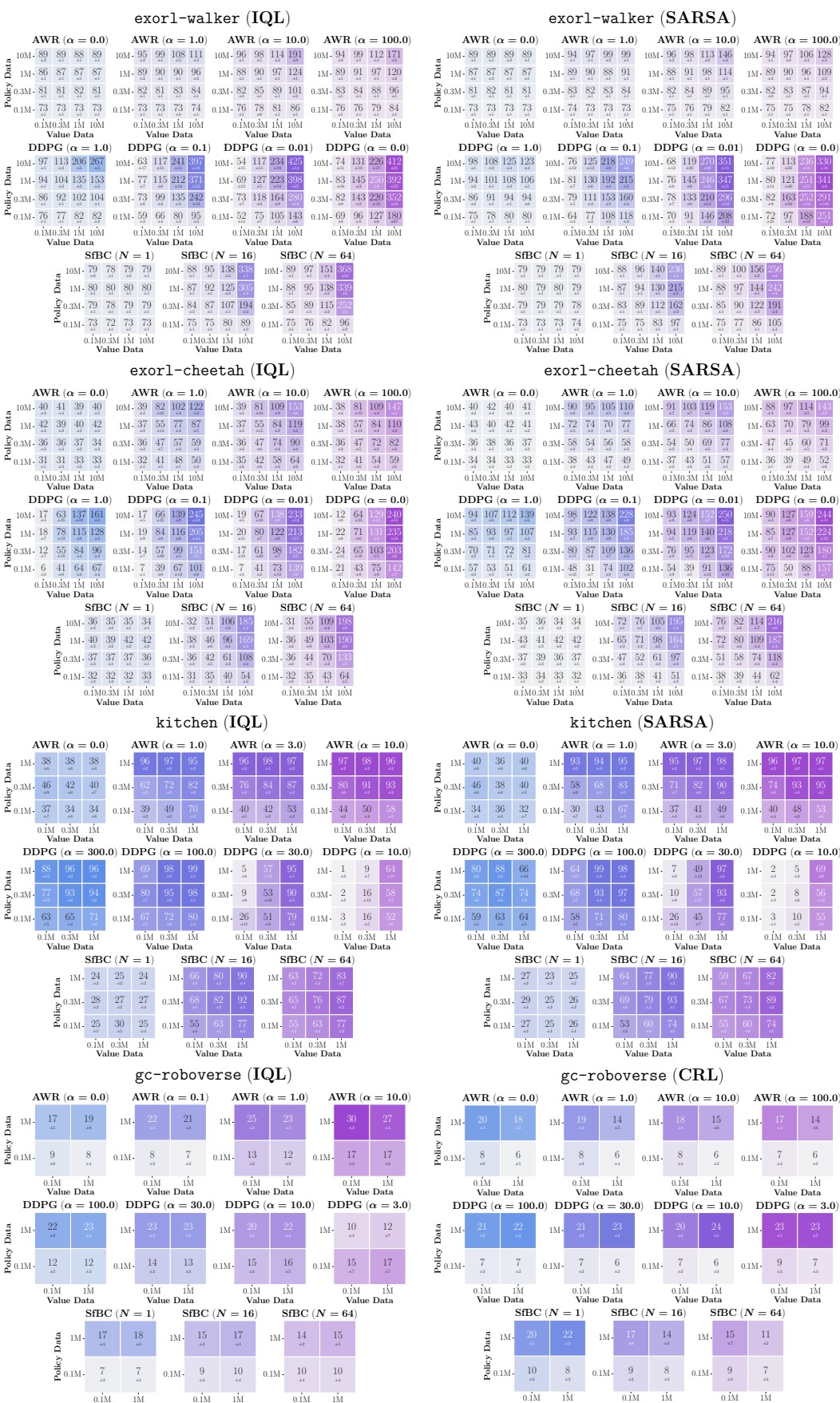

Figure 11: **Full data-scaling matrices of AWR, DDPG+BC, and SfBC with different hyperparameters.**

