# OpenReview forum: "Is Value Learning Really the Main Bottleneck in Offline RL?"
_NeurIPS.cc/2024/Conference — NeurIPS 2024 poster_

### Official Review · Reviewer_VnDz · 2024-06-30

**Soundness:** 3
**Presentation:** 4
**Contribution:** 3
**Rating:** 7
**Confidence:** 4

**Summary:**

The paper presents an empirical analysis to determine the main challenge in offline RL for control among value function learning, policy extraction, and policy generalization to test-time states. With various deep learning-based experiments, it reaches the conclusion that policy extraction and policy generalization are the main bottlenecks instead of value function learning.

**Strengths:**

1. The paper is systematic on specifically isolating the three components such as by using decoupled RL algorithms (having value function learning and policy learning phase separately). This is important to avoid confounding factors.
2. The paper covers results across various data dimensions such as coverage, sub-optimality, and amount.
3. The paper has clear takeaways, which are helpful in determining actionable advice.
4. The paper provides an alternative direction (e.g. better policy extraction algorithms instead of value function learning algorithms)  for researchers to pursue in trying to improve offline RL.

**Weaknesses:**

1. It is a bit concerning that the main results in Figures 1, 2, 6 are over only 4 seeds. In my experience, offline RL algorithms can be quite fragile especially with so much variation in coverage, dataset amount etc, that 4 seems quite limited. Given that most of the environments are not image-based, more than 4 seeds seems reasonable.
2. Related to above, the same figures do not report any information on variance of performance, which makes it difficult to determine if the reported means have been accurately reported. I would suggest reviewing some guidelines [1]
3. While the takeaways are nice, the claim to “always use” (above Section 5) is too strong, in my opinion.
4. In section 4.4, it's unclear if the first reason is a good explanation for better performance. While the actions of DDPG + BC are more extreme, I don't think extreme actions are what we should be aiming for. The experiments are presented as though these extreme actions lead to good performance (even if this was not the intention). While it may be true in the evaluated domains, it is unclear if this can be a general finding.

I will be willing to increase the score if the issues above are addressed, especially 1 and 2.

[1] Empirical Design in Reinforcement Learning. Patterson et al. 2023.

**Questions:**

1. There seems like there is a relation between DDPG + BC’s focus on limited exploration (some improvement but staying close to behavior policy) and test-time generalization. More specifically, by encouraging a policy to be close to the behavior policy, the test-time generalization should naturally be better since it will not deviate much from those states. Section 4 does not comment on this relation, what are the authors thoughts on this?
2. Regarding the limitations mentioned at the end, the authors may be interested in this paper [1] as well which discusses how some value-based objectives may be unreliable for performance. [1] does not give an suggestions, but does report a similar finding for another use-case.

[1] Why Should I Trust You, Bellman? The Bellman Error is a Poor Replacement for Value Error. Fujimoto et al. 2022.

**Limitations:**

Yes the authors report on the limitations in Appendix A.

---

> ### Author Rebuttal · Authors · 2024-08-05
>
> Thank you for the detailed review and constructive feedback about this work. We especially appreciate the reviewer's feedback about statistical significance and the clarity of our claims. Following the reviewer’s suggestion, we have added more seeds and variance metrics to improve the statistical significance of the results. We believe these changes have strengthened the paper substantially. Please find our detailed responses below.
>
> ---
>
> * **“It is a bit concerning that the main results in Figures 1, 2, 6 are over only 4 seeds.” / “the same figures do not report any information on variance of performance”**
>
> Thanks for raising this point. Following the suggestion, we’ve added 4 more seeds to Figures 1, 2, and 6 by adding **7744 more runs** (now we have **8 seeds** in total for every experiment in the paper) and have reported **standard deviation metrics** in our data-scaling matrices. Please find the improved results in **Figure 1 in the additional 1-page PDF**. Our conclusions remain the same, and we have updated these results in the current version of the paper.
>
> * **About extreme actions**
>
> We fully agree with the reviewer that extreme actions do not necessarily mean they are more optimal. Our original intention was to show that AWR has *limited expressivity* compared to DDPG+BC, because AWR actions *always* lie in the convex hull of the dataset actions (at least in principle), while DDPG+BC can go outside of it. This resulted in more extreme actions for DDPG+BC (Figure 3), but this is not to say that extreme actions are better. We apologize for the confusion and we will clarify this point in the draft to prevent such potential confusion.
>
> * **“the claim to “always use” (above Section 5) is too strong”**
>
> Thanks for the suggestion. We initially used the term “always” because we found that DDPG+BC is better than or as good as AWR in most of the cases (15 out of 16 settings — please refer to **Table 1 in the new 1-page PDF**). However, following the suggestion, we have toned down the takeaway by removing the word “always” in the current draft.
>
> * **“DDPG+BC’s focus on limited exploration” / “by encouraging a policy to be close to the behavior policy, the test-time generalization should naturally be better since it will not deviate much from those states”**
>
> Thanks for the great question. We also believe that, in general, the higher the BC coefficient in DDPG+BC is, the better it can prevent encountering out-of-distribution states at test time. (We’d also like to note that this also applies to AWR since it has a temperature hyperparameter of a similar role.) However, in practice, due to function approximation or imperfect policy learning, the agent would almost always visit out-of-distribution states, even when there’s *only* the BC term, and thus the generalizability of the policy can still significantly affect performance even in this case. We provide one example of such evidence in Appendix C. That said, as the reviewer mentioned, we expect that a small BC coefficient would incur even further challenges in generalization.
>
> * **About the Fujimoto et al. paper**
>
> Thanks for the pointer! We think this paper indeed points out similar limitations of the Bellman error. We will cite and discuss this paper.
>
> We would like to thank the reviewer again for raising important points about statistical significance and clarity, and please let us know if there are any additional concerns or questions.

---

> > ### Comment · Reviewer_VnDz · 2024-08-07
> >
> > Thanks to the authors for addressing my concerns and running more trials! I have updated my score.

---

> > > ### Author Response · Authors · 2024-08-12
> > > **Official Comment by Authors**
> > >
> > > We would like to thank the reviewer for appreciating our changes and adjusting the score accordingly. We believe these updates and clarifications have indeed strengthened the paper.

---

### Official Review · Reviewer_C9VZ · 2024-07-07

**Soundness:** 2
**Presentation:** 2
**Contribution:** 2
**Rating:** 6
**Confidence:** 4

**Summary:**

This paper attempts to understand the relative importance of policy learning and value learning in offline reinforcement learning. The analysis is broken into two parts: (1) when decoupling the policy and value learning steps the authors test the relative data efficiency of the two steps, and (2) the authors test how well the policy generalizes at test time. Experiments are conducted on a wide variety of domains and the authors argue that they show that policy learning is often the main bottleneck of offline RL.

**Strengths:**

1. The high level experimental methodology of comparing the relative importance of policy learning and value learning by varying the dataset size in decoupled algorithms is an interesting idea. It could be useful to guide future work to know if there is more upside to improving on policy learning or value learning.

2. The result that IQL+DDPG outperforms IQL+AWR (which was used in the original IQL paper) is an interesting standalone result.

3. While somewhat preliminary and cursory in the paper, the idea of OPEX and TTT to update the policy at test time with a fixed value is interesting as a way to show that the value function has more information in it than the policy extracts.

**Weaknesses:**

1. The results are very heuristic and often unclear. The definitions of "policy bound" and "value bound" in terms of color gradients is vague and not very informative. Visually, it is hard to get much out of the figures which just throw a ton of data at the reader without a very intuitive way to interpret it. This could maybe be resolved by creating a more clear single metric that indicates the tradeoffs or some summary statistics that show aggregate trends averaging across methods or datasets. Currently, the results are pretty hard to parse and not very convincing as a result.

2. The empirical results are not as conclusive as the analysis/text suggests. For example, the main claim of section 4 that "policy learning is often the main bottleneck of offline RL" does not seem to be the right takeaway or emphasis of the results. Instead the results in both figure 1 and 2 indicate that sometimes policy learning is the bottleneck and sometimes value learning is the bottleneck.

3. The methodology in section 5 is a not clear. It seems that the authors run offline-to-online IQL (line 304), but this would be using AWR, which the previous section suggests not to do. Moreover, the online algorithm updates not just the policy, but also the value. The paper does not consider the generalization of the value function, but only the policy. Perhaps a cleaner way to test interesting out of distribution generalization would be to consider the state distribution of the optimal policy? This would of course not test near optimal states (so maybe some noise could be added or something as in later experiments), but could be conceptually cleaner than fully changing the setting to online.

4. It is not clear why there is such focus on the generalization of the policies and not the values. They seem to both matter, especially for things like OPEX or TTT to work. It would be interesting to see how well the value functions are generalizing as well as the policies to go to the papers main claims to be comparing the relative importance of these two steps.

5. In general, the paper tries to cram in too many semi-related results. I would encourage the authors to focus the story a bit more clearly and maybe split some parts into the appendix or into a separate paper.

**Questions:**

See weaknesses

**Limitations:**

Yes, but could do more to address how the results are not always clear cut.

---

> ### Author Rebuttal · Authors · 2024-08-05
>
> Thank you for the detailed review and constructive feedback about this work. It appears that we were not entirely clear about our main messages in the initial draft, which we believe may have caused some confusion about our claims. Below we describe how we have revised our paper to prevent potential confusion and have added new results about aggregation metrics and generalization analysis.
>
> ---
> * **The results are not as conclusive as the paper suggests; they indicate that sometimes policy learning is the bottleneck and sometimes value learning is the bottleneck.**
>
> Thank you for raising this valid point. We believe our initial draft was not entirely clear about our analysis framework. We first would like to clarify that there are *two* types of information we can obtain from the data-scaling matrices (Figure 1).
> - (1) By looking at each **individual matrix**, we can see scaling behaviors with the amounts of value and policy data in that *specific* setting.
> - (2) By comparing different matrices from **different value/policy algorithms**, we can understand the effect of each algorithmic component.
>
> As the reviewer pointed out, if we look at individual matrices (the first perspective), some matrices are value-bounded (i.e., the performances are more affected by the amount of value data) and others are policy-bounded, and the results appear to be less clear and problem-dependent. **However, if we compare different algorithms (the second perspective), we can observe a much clearer trend in our results:** namely, (1) the choice of a policy extraction algorithm often affects performance more than the choice of a value learning algorithm (except antmaze-large), and (2) DDPG+BC is almost always better than AWR. Please refer to our **global response** for aggregation metrics that quantitatively support these observations.
>
> We then find the reason behind this difference between AWR and DDPG+BC **by now looking at individual matrices** (Section 4.3). The gc-antmaze result in Figure 2 shows a clear difference between the two algorithms: with AWR, an increase in value data doesn’t necessarily translate to performance improvement, unlike with DDPG+BC. This suggests that AWR, one of the most widely used policy extraction algorithms, imposes a **“bottleneck”** that inhibits the full use of the learned value function (the “Takeaway” box in Section 4). In this sense, we argue that policy extraction is often the main bottleneck in current offline RL algorithms (esp. any method that involves weighted/filtered BC, including the recent scaling work [4]). Please note that this is different from saying that every *individual* data-scaling matrix is policy-bounded (which is indeed not true).
>
> We believe our initial draft was not very clear about these points, and will revise the manuscript to clearly convey these findings.
>
> * **“the results are pretty hard to parse” / “Lack of clear aggregation metrics”**
>
> Thank you for the helpful suggestion. Following the suggestion, we have added two types of quantitative aggregation metrics. Please refer to our **global response** for the details.
>
> * **“It is not clear why there is such focus on the generalization of the policies and not the values.”**
>
> Thanks for raising this question. First, we would like to note that our main focus in the second analysis (Section 5) is **not** on comparing value generalization and policy generalization (unlike our first analysis); our main point is rather to show that *test-time generalization* (regardless of policy or value) is one of the important yet overlooked issues in offline RL. We used the phrase “policy generalization” in the paper because the policy is what is deployed at test time, but we didn’t intend to mean that policy generalization is necessarily more important than value generalization. Instead, we wanted to show how current offline RL algorithms are often *already* great at learning near-optimal actions on in-distribution states, and how bad they can be on test-time out-of-distribution states. This observation highlights an important (but often overlooked) open question in offline RL, which is very different from the previous main focus on "value learning" about pessimism and conservatism. We apologize for this potential confusion (which we think stems from some phrases in Sections 1, 3, and 5), and will revise the paper to make this point very clear (e.g., we will use just “generalization” not “policy generalization” whenever it’s more appropriate).
>
> That said, we do have some empirical results that allow us to compare policy and value generalization, and we discuss these aspects in the **global response** above in case it’s of interest to the reviewer.
>
> * **The use of IQL+AWR in our offline-to-online RL experiment / MSE under** $d^{\pi^*}$
>
> Thanks for the suggestion. Following the suggestion, we have repeated the same experiment with IQL+DDPG+BC and additionally measured the MSE under $d^{\pi^*}$. We obtained similar results and our conclusion remains the same. Please refer to our **global response** for the details.
>
> * **“In general, the paper tries to cram in too many semi-related results.”**
>
> Our primary motivation in this work is to understand the best way to improve the performance of current offline RL. To this end, we believe conducting a **holistic** analysis of different elements that affect the performance of offline RL is necessary, and thus we carried out such analyses (across value learning, policy extraction, and generalization) in this paper. That being said, to provide more coherent insights, we have substantially revised several subsections and paragraphs in the current version, which we hope helps address this concern.
>
> We would like to thank the reviewer again for raising important points about our claims, which substantially helped strengthen the paper. **Please let us know if we have addressed the reviewer’s concerns and if so, we would be grateful if you are willing to upgrade the score.**

---

> > ### Comment · Reviewer_C9VZ · 2024-08-10
> >
> > Thanks for the thorough response and additional experiments/plots.
> >
> > - Indeed, aggregating the metrics does provide significantly more evidence for the main claim of the paper. I would highly recommend making these the main results figures and using the huge block of heatmaps as supporting evidence in the appendix. One other small suggestion would be to add IQM metrics that aggregate the value learning methods *only* across the best policy extraction method per-environment, and the policy learning methods *only* across the best value learning method. While this is using a sort of "oracle" for the other part of the algorithm, it would be nice to see aggregates that are not biased by averaging in the worst choices for the other half of the algorithm.
> >
> > - And ok, I see that you want the second half of the paper to be about generalization of both policy and value in some sense. If this is true, I would suggest reframing the title/abstract/intro to reflect that this is a distinct point from the policy vs. value comparison. Or, as suggested in the initial review, maybe this make more sense as two separate papers, the connection is still not super clear to me.
> >
> > I will increase my score to a 6 to reflect the clearer results and framing, but still with some uncertainty since the proposed changes are substantial and the two halves of the paper do not seem to quite hang together yet.

---

> > > ### Author Response · Authors · 2024-08-12
> > > **Official Comment by Authors**
> > >
> > > Thanks for engaging in the discussion and adjusting the score accordingly. We are grateful that the reviewer appreciates the changes, and we will update the paper to incorporate the proposed changes and additional results.
> > >
> > > With regards to the question that the "two halves of the paper do not seem to quite hang together", we would like to articulate our rationale for studying policy extraction and generalization together in this paper more clearly. Akin to how with any ML algorithm, there are two problems: optimization and generalization (e.g., an ERM bound decomposes into two terms: one focusing on optimization and the other on generalization), the performance of any RL algorithm is also affected by the efficacy of policy and value optimization, and corresponding generalization. If the policy and value functions were optimized perfectly and could generalize perfectly as well, then that offline RL algorithm should attain perfect performance. Conversely, if any of these components do not function as well, the performance would not be perfect. Therefore, to be able to perform a holistic analysis of offline RL challenges, we separated them into (1) value and policy learning, and (2) generalization.
> > >
> > > While it might seem approaches to analyze and improve policy extraction, value learning, and generalization might look distinct from each other, as the reviewer mentioned, we would like to note that our main goal is to exhaustively highlight the challenges/bottlenecks in an existing research area. This is analogous to how several prior analysis papers in RL also present a diverse set of challenges and propose methods that might appear disconnected -- for example, Fu et al. (2019) [1] studied sampling errors (high UTD), replay buffers, function approximation, and sampling schemes -- all in one paper, without much connection to each other necessarily, but have influenced many follow-up works on individual topics such as Q-function divergence, replay buffer studies, sampling distributions, etc; Lyle et al. (2023) [2] studied the plasticity loss phenomenon from a variety of perspectives, across from supervised learning to RL and from optimizers to metrics and solutions, where these various insights have motivated various subsequent works and solutions. Likewise, we hope that our analysis results, across the three bottlenecks in offline RL, motivate future work into building techniques to solve each of these challenges, perhaps from the starting points shown in various sections of our paper, resulting in more advanced and effective offline RL algorithms.
> > >
> > > Once again, we would like to thank the reviewer for the suggestions, which we believe significantly helped improve the quality of the paper.
> > >
> > > [1] Fu et al., Diagnosing bottlenecks in Deep Q-Learning Algorithms. ICML 2019. \
> > > [2] Lyle et al., Understanding plasticity in neural networks, ICML 2023.

---

### Official Review · Reviewer_Bu3n · 2024-07-10

**Soundness:** 3
**Presentation:** 3
**Contribution:** 3
**Rating:** 7
**Confidence:** 4

**Summary:**

This paper empirically analyzes the bottlenecks in offline RL from three aspects: value learning, policy extraction, and policy generalization at evaluation time. Through the empirical evaluation, two observations were made: 1) the policy extraction algorithms affect the performance of offline RL significantly, often more than its underlying value learning objective. 2) the sub-optimal performance of offline RL agents is mainly attributed to a lack of generalization of the unseen state during evaluation instead of accuracy in the training distribution.

**Strengths:**

**Originality**: Good. This paper tries to analyze the bottleneck of offline RL methods and includes some novel and interesting observations that were not systematically discussed before.

**Clarity**: Good. The paper is well-structured and easy to follow. The takeaway after the empirical analysis helps the reader understand the experiment's results.

**Significance**: This is important. The reason behind offline RL's underperformance is an important topic for the future development of offline RL methods.

**Technical Accuracy**:

1. Throughout evaluation with a decent amount of experiments
2. The experiment design are motivated and backed by hypothesizes

**Weaknesses:**

1. There is a lack of variance measures, such as standard deviation (std) or confidence intervals (CI), for the data-scaling matrices in Figures 1, 2, and 6, making the numerical results less plausible.

2. For some experiments in empirical analysis 1, an increase in data leads to a decrease in performance. For instance, in Figure 1 (gc-antmaze-large), both IQL+AWR and IQL+DDPG show that the 10-10 configuration performs worse than the 1-10 and 10-1 configurations, which seems to contradict to the assumption, more data leads to better value/policy approximation. These observations lack an adequate explanation.

3. I am concern that one of the main question (as outlined in the title as well) “is value function learning the main bottleneck of offline RL” is not sufficiently examined in the empirical experiments. While the amount of data used for training can be an implicit indicator of how well the value function is trained, it does not directly tell us the distribution of approximation errors (e.g., overestimation). Overestimation could still be a significant issue in offline reinforcement learning. It would strengthen the augment if an direct comparison between the predict value and true value for can be conducted, just like in [1] and [2].

[1] Van Hasselt, H., Guez, A. and Silver, D., 2016, March. Deep reinforcement learning with double q-learning. In *Proceedings of the AAAI conference on artificial intelligence* (Vol. 30, No. 1).

[2] Fujimoto, S., Hoof, H. and Meger, D., 2018, July. Addressing function approximation error in actor-critic methods. In *International conference on machine learning* (pp. 1587-1596). PMLR.

**Questions:**

1. For empirical analysis 2, I am wondering if the online data mainly improves value learning, policy learning, or both of them. The current evaluation only shows the improvement of evaluation MSE in actions but I'm wondering if the Q-value estimation loss would follow the same pattern, maybe the author could provide more insights on this?

**Limitations:**

Limitations were not discussed in the paper.

---

> ### Author Rebuttal · Authors · 2024-08-05
>
> Thank you for the detailed review and constructive feedback about this work. We especially appreciate the reviewer's feedback about statistical significance as well as the question on our claim about value learning. Following the reviewer’s suggestion, we have added variance metrics as well as four more seeds (8 seeds in total) to improve the statistical significance of our results. We believe these changes have strengthened the paper substantially. Please find our detailed answers below.
>
> ---
>
> * **“lack of variance measures”**
>
> Thanks for raising this point. Following the suggestion, we have added standard deviations to our data-scaling matrices, and report them in **Figure 1 in the additional 1-page PDF**. We have also updated our paper accordingly.
>
> * **“For some experiments in empirical analysis 1, an increase in data leads to a decrease in performance.”**
>
> As the reviewer mentioned, there are some cases where an increase in data doesn’t necessarily lead to improved performance. We believe this is mainly due to statistical errors. To address this concern, (1) we’ve added 4 more seeds to our data-scaling matrices by adding **7744 more runs** (now we have **8 seeds** in total for every experiment in the paper), and (2) we have added variance metrics to the table. Please find them in **Figure 1 in the new 1-page PDF**. There are still some rare cases where an increase in data doesn't lead to performance improvement, but the new standard deviation metrics tell us that this is likely because of statistical noise. We believe that these changes have significantly strengthened the statistical significance of our results.
>
> * **About the title / overestimation in value learning**
>
> We agree that overestimation is an important issue in offline value learning in general, and would like to clarify the scope of our claims. Concretely, we would like to clarify that we’re *not* suggesting that value learning is less or not important compared to policy learning in general. Rather, our claim is that, (due to recent advancements in offline value function training) *current* state-of-the-art offline RL methods often tend to be at the point where improvements in policy extraction and generalization can lead to larger improvements in performance; in this sense, we claim that policy extraction and generalization are often the main “bottlenecks” in current offline RL, hence the title. As we show in our experiments (Figures 1 and 2), we are at the point where improvements in value functions (by adding more data) often do not translate to policy performance if the policy learning algorithm is not chosen appropriately. Hence, we think it is useful to concretely highlight and emphasize policy learning as a bottleneck at this point since the community has focused on value learning substantially. Thanks for asking this question, and we have revised our paper (Intro and Section 3) to make this point clearer.
>
> * **“For empirical analysis 2, I am wondering if the online data mainly improves value learning, policy learning, or both of them.”**
>
> Thanks for the interesting question. In this paper, we mainly measure the accuracy of the policy because the policy is what is deployed at test time. We expect additional online data would improve both the value and the policy in general, but it might be a bit challenging to faithfully measure the accuracy of the learned value function (as opposed to policy accuracy), given that (1) the Bellman loss often does not necessarily correlate with the actual performance [3] and (2) offline value functions (especially with techniques for mitigating overestimation like IQL/CQL) often involve pessimism/conservatism, which makes it difficult to directly compare against the actual ground-truth Q value. While we do have a comparison between the value and the policy in terms of data-scaling in the paper, we leave dissecting the effect of each component for *generalization* (with additional online data) for future work.
>
>
> We would like to thank the reviewer again for raising important points about statistical significance and the score of our claims, and please let us know if there are any additional concerns or questions.

---

> > ### Comment · Reviewer_Bu3n · 2024-08-08
> >
> > Thank the authors for their detailed responses and extra experiments. The statistical measurements make the results more convincing. I will keep my accepting rating.

---

> ### Author Response · Authors · 2024-08-12
> **Official Comment by Authors**
>
> We would like to thank the reviewer for appreciating our changes. We believe these updates and clarifications have indeed strengthened the paper.

---

### Official Review · Reviewer_BVrx · 2024-07-23

**Soundness:** 4
**Presentation:** 4
**Contribution:** 3
**Rating:** 8
**Confidence:** 4

**Summary:**

This paper addresses the question of why offline RL often underperforms imitation learning. They formalize the question they choose to ask, " is the bottleneck in learning the value function, the policy, or something else? What is the best way to improve performance given the bottleneck?", and provide three potential explanations: imperfect value function estimation, imperfect, policy extraction from value function, and imperfect policy generalization to novel states during evaluation. They next perform a series of experiments to test each hypothesis, and conclude two main reasons for offline RL's underperformance: policy extraction from value functions, and test-time policy generalization. They use these observations to highlight important takeaways for RL practitioner and RL researchers.

**Strengths:**

I think that this is an exceptionally well-written paper. They make it very clear what their hypotheses are, how they test for each, and what the reader should take away from each subsection.

I further think that this paper addresses important questions in offline RL, and brings to light important directions for future work, which is a valuable contribution.

The full experimental details as well as code are provided, which is very useful for the community in reproducing the results and building off of this work.

**Weaknesses:**

At a high level, there are not many weaknesses I can name in this work. One is perhaps that their contributions are mainly empirical observations, and it would be nice to support these with theoretical results (even in very simple settings full of necessary assumptions), but I believe that even without theory this paper is very strong.

**Questions:**

From what I can tell, most of the empirical results are in environments with continuous action spaces. Do you expect the results to carry over to environments with discrete action spaces?

**Limitations:**

The authors discuss and address limitations of their work.

---

> ### Author Rebuttal · Authors · 2024-08-05
>
> Thank you for the positive review and constructive feedback about this work! We especially appreciate the question on discrete-action MDPs. Please find our answer to the question below.
>
> ---
>
> * **“... most of the empirical results are in environments with continuous action spaces. Do you expect the results to carry over to environments with discrete action spaces?”**
>
> Thanks for raising this point. There are two main differences between discrete-action and continuous-action MDPs: (1) discrete-action MDPs do not always require a separate policy extraction procedure, as we can directly enumerate over actions to choose the argmax action, and (2) DDPG is not straightforwardly defined in discrete-action MDPs. Hence, one of our takeaway messages (“use DDPG+BC instead of AWR”) may not be directly applicable to discrete-action MDPs. However, we expect that our main findings — namely (1) policy extraction can inhibit the full use of the learned value function (if there’s a separate policy extraction step) and (2) test-time policy generalization is one of the significant bottlenecks in offline RL — still apply to discrete-action MDPs as well. For example, recent works in RL fine-tuning of LLMs, which use a discrete action space, have observed similar findings to us: Tajwar et al. (2024) [1] showed that sampling actions (i.e., responses) from the policy (akin to sampling actions from the policy for making an update in DDPG+BC) lead to better performance than AWR that does not sample a new action from the policy for training, and Cobbe et al. [2] showed the effectiveness of test-time verifier reranking methods, which is closely related to the test-time generalization argument and TTT/OPEX in our paper. While of course we do not focus on LLMs in this study, these prior works illustrate that similar findings are applicable to discrete actions. We will add a discussion about discrete-action MDPs in the paper.
>
> We would like to thank the reviewer again for the helpful feedback and please let us know if there are any additional concerns or questions.

---

### Author Rebuttal · Authors · 2024-08-05

We appreciate all four reviewers’ detailed feedback and suggestions. We would like to highlight the additional results we provide in the new 1-page PDF.

* **Adding $\mathbf{4}$ more seeds ($\mathbf{8}$ seeds in total) and standard deviation metrics:** Following the reviewers’ suggestions, we have added $\mathbf{4}$ **more seeds** as well as **standard deviation metrics** to our data-scaling matrices by adding $\mathbf{7744}$ **more runs** (**Figure 1** in the new PDF). Now, every experiment in our paper is aggregated over $\mathbf{8}$ **seeds** in total.
* **Aggregation metrics:** Following the suggestion of Reviewer C9VZ, we have added two new aggregation metrics that clearly highlight our takeaways.
  * First, we aggregate normalized returns over the entire data-scaling matrices for each value and policy algorithm (by marginalizing over every other factor), and report the interquartile mean (IQM) metrics [5] in **Figure 2** in the new PDF. The results support our finding that the difference between policy extraction methods is often larger than that between value learning methods.
  * Second, we aggregate the performances from different policy extraction methods for each environment and value algorithm. **Table 1** in the new PDF shows that DDPG+BC is indeed almost always (15 out of 16 tasks) better than or as good as AWR. We will add these results to the paper.
* **Offline-to-online RL experiment with IQL+DDPG+BC:** To address a concern of Reviewer C9VZ, we have repeated our offline-to-online RL experiment (Figure 5 in the original draft) with IQL+DDPG+BC, instead of IQL+AWR.
  * In the original offline-to-online RL experiment, we used IQL+AWR because (1) we wanted to use the simplest setting to illustrate the generalization issue (note that DDPG+BC requires a separate exploration mechanism in the online phase since it learns a deterministic policy, unlike AWR), and (2) most of the pathologies of AWR (e.g., the mode-covering issue, limited expressivity, etc.) go away if we’re allowed to use *online* rollouts [1].
  * That being said, following the suggestion of Reviewer C9VZ, we repeated the same offline-to-online experiments with IQL+DDPG+BC with Gaussian exploration noises, and report the results in **Figure 3** in the new PDF. The new results show a very similar trend to the original results, suggesting that the conclusion remains the same.
  * In addition, following the suggestion of the reviewer, we also measure the MSE metric under the state-marginal distribution of the oracle optimal policy (“$d^{\pi^*}$ MSE”). **Figure 3** in the new PDF shows the results, which suggests that the $d^{\pi^*}$ MSE is also correlated with performance (especially compared to the training/validation MSEs), but the correlation is slightly weaker than the evaluation MSE metric. We believe this is because the evaluation MSE directly measures the accuracy of the current policy under the *current* policy distribution.

---

Additionally, below we provide additional comments about test-time generalization to **Reviewer C9VZ**. We provide them here due to the character limit of each response.

* **Additional comments to Reviewer C9VZ about the point “It is not clear why there is such focus on the generalization of the policies and not the values.”**

As we discussed in the main response, our main focus in the second analysis is *not* to argue that policy generalization is necessarily more important than value generalization, but to highlight the significance of the effect of test-time generalization (regardless of policy or value) on performance, which is an important but often overlooked bottleneck in offline RL. That being said, in the paper, we do have some empirical results that allow us to compare policy generalization and value generalization, and we discuss them here in case it is of interest to the reviewer.

First, as the reviewer pointed out, our experiments with OPEX/TTT imply that *values often generalize better than policies*. OPEX/TTT only updates the policy (or actions) from the *fixed* offline value function on test-time states, and the fact that this often improves performance indicates that the information in the value function has often not been fully transferred into the policy. Hence, in this case, we would be able to say that *policy* generalization is being the “bottleneck” in performance.

Second, we also found that policy generalization *alone* can affect performance significantly. The results in Appendix C suggest this, where we show that just changing the representation of the BC policy can lead to a very significant difference in performance (this in fact leads to near-SOTA performance on gc-antmaze-large, even without using any RL!) and we show that this is due to nothing other than the difference in policy generalizability (note that this is a BC setting, so there’s no value function).

---

Below are the references that we use throughout our response:

[1] Tajwar et al., Preference Fine-Tuning of LLMs Should Leverage Suboptimal, On-Policy Data, ICML 2024. \
[2] Cobbe et al., Training Verifiers to Solve Math Word Problems, 2021. \
[3] Fujimoto et al., ​​Why Should I Trust You, Bellman? The Bellman Error is a Poor Replacement for Value Error, ICML 2022. \
[4] Springenberg et al., Offline Actor-Critic Reinforcement Learning Scales to Large Models, ICML 2024. \
[5] Agarwal et al., Deep Reinforcement Learning at the Edge of the Statistical Precipice, NeurIPS 2021.

---

### Decision · Program_Chairs · 2024-09-25

**Decision:**

Accept (poster)

**Comment:**

While numerous issues were raised at the initial review stage, clarifications and additional experimentation during the rebuttal convinced all four reviewers that the paper merits acceptance.  The authors should incorporate clarifications and new experimental results from the rebuttal into the final version of their paper.